# Ca²⁺-dependent release of synaptotagmin-1 from the SNARE complex on phosphatidylinositol 4,5-bisphosphate-containing membranes

Rashmi Voleti[1,2,3†], Klaudia Jaczynska[1,2,3†], Josep Rizo[1,2,3*]

[1]Department of Biophysics, University of Texas Southwestern Medical Center, Dallas, United States; [2]Department of Biochemistry, University of Texas Southwestern Medical Center, Dallas, United States; [3]Department of Pharmacology, University of Texas Southwestern Medical Center, Dallas, United States

**Abstract** The Ca²⁺ sensor synaptotagmin-1 and the SNARE complex cooperate to trigger neurotransmitter release. Structural studies elucidated three distinct synaptotagmin-1-SNARE complex binding modes involving 'polybasic', 'primary' and 'tripartite' interfaces of synaptotagmin-1. We investigated these interactions using NMR and fluorescence spectroscopy. Synaptotagmin-1 binds to the SNARE complex through the polybasic and primary interfaces in solution. Ca²⁺-free synaptotagmin-1 binds to SNARE complexes anchored on PIP₂-containing nanodiscs. R398Q/ R399Q and E295A/Y338W mutations at the primary interface, which strongly impair neurotransmitter release, disrupt and enhance synaptotagmin-1-SNARE complex binding, respectively. Ca²⁺ induces tight binding of synaptotagmin-1 to PIP₂-containing nanodiscs, disrupting synaptotagmin-1-SNARE interactions. Specific effects of mutations in the polybasic region on Ca²⁺-dependent synaptotagmin-1-PIP₂-membrane interactions correlate with their effects on release. Our data suggest that synaptotagmin-1 binds to the SNARE complex through the primary interface and that Ca²⁺ releases this interaction, inducing PIP₂/membrane binding and allowing cooperation between synaptotagmin-1 and the SNAREs in membrane fusion to trigger release.

*For correspondence:
Jose.Rizo-Rey@UTSouthwestern.
edu

†These authors contributed
equally to this work

Competing interests: The
authors declare that no
competing interests exist.

Reviewing editor: Nils Brose,
Max Planck Institute of
Experimental Medicine, Germany

## Introduction

The release of neurotransmitters by Ca²⁺-evoked synaptic vesicle exocytosis is an exquisitely regulated process that is critical for communication between neurons. Release involves tethering of synaptic vesicles to presynaptic active zones, priming of the vesicles to a release-ready state, and Ca²⁺-triggered fusion of the vesicle and plasma membranes (*Südhof, 2013*). Extensive research has allowed reconstitution of basic features of synaptic vesicle fusion with the central components of the neurotransmitter release machinery (*Liu et al., 2016*; *Ma et al., 2013*) and led to defined models for their functions (*Brunger et al., 2018*; *Rizo, 2018*). The SNAP receptors (SNAREs) syntaxin-1, SNAP-25 and synaptobrevin form a tight four-helix bundle called the SNARE complex that brings the vesicle and plasma membranes together and is key for membrane fusion (*Hanson et al., 1997*; *Poirier et al., 1998*; *Söllner et al., 1993*; *Sutton et al., 1998*). This complex is disassembled by N-ethylmaleimide sensitive factor (NSF) and soluble NSF adaptors proteins (SNAPs) (*Söllner et al., 1993*), whereas SNARE complex assembly is organized in an NSF-SNAP-resistant manner by Munc18-1 and Munc13s (*Ma et al., 2013*; *Prinslow et al., 2019*). Release is triggered very fast (<1 ms) after Ca²⁺ influx by the Ca²⁺ sensor Synaptotagmin-1 (Syt1) (*Fernández-Chacón et al., 2001*).

**eLife digest** Inside the brain, cells called neurons relay messages from one place to another in the form of electrical signals. When an electrical signal reaches a junction between two neurons (known as a synapse) it triggers small particles called calcium ions to enter one of the cells. This influx of calcium causes vesicles to fuse with the membrane surrounding the neuron and release molecules called neurotransmitters into the small gap between the two neurons. These molecules travel across the gap to activate an electrical signal in the second neuron which then carries the message onwards.

A protein known as synaptotagmin-1 senses calcium ions at synapses and works together with a group of proteins known as the SNARE complex to help vesicles fuse with the cell membrane. Previous studies have reported three different structures of synaptotagmin-1 bound to the SNARE complex in a different way. But it was unclear which of these binding states actually result in the release of neurotransmitters. To address this question, Voleti, Jaczynska and Rizo studied how and when synptotagmin-1 and the SNARE complex bind together using two approaches known as NMR spectroscopy and fluorescence spectroscopy.

The experiments suggest that before calcium enters the synapse, synaptotagmin-1 is already bound to a surface on the SNARE complex. This binding inhibits the release of neurotransmitters and has been reported in previous studies. Adding calcium ions causes synaptotagmin-1 to be released from the SNARE complex. This allows synaptotagmin-1 to interact with the membrane and cooperate with the SNARE complex to trigger vesicle fusion.

Finding out how neurons release neurotransmitters at synapses may help us to understand how the brain works. This could provide new insights into how defects in the synapse lead to neurological disorders, such as schizophrenia, and potentially aid the development of new treatments for such conditions.

Syt1 is a synaptic vesicle protein with two $C_2$ domains (named $C_2A$ and $C_2B$) that form most of its cytoplasmic region and adopt β-sandwich structures that bind multiple $Ca^{2+}$ ions through loops at the top of the β-sandwich (*Fernandez et al., 2001*; *Sutton et al., 1995*; *Ubach et al., 1998*). These loops also mediate $Ca^{2+}$-dependent binding to phospholipids (*Chapman and Davis, 1998*; *Frazier et al., 2003*; *Zhang et al., 1998*), which is crucial for neurotransmitter release (*Fernández-Chacón et al., 2001*; *Rhee et al., 2005*). A polybasic region on the side of the $C_2B$ domain β-sandwich also contributes to membrane binding, in part via interactions with phosphatidylinositol 4,5-bisphosphate (PIP$_2$) (*Bai et al., 2004*; *Li et al., 2006*). In addition, the Syt1 $C_2B$ domain can bind simultaneously to two membranes in a $Ca^{2+}$ dependent manner through its $Ca^{2+}$-binding loops and two arginines at the bottom of the β-sandwich (R398,R399) that are crucial for Syt1 function (*Xue et al., 2008*), suggesting that Syt1 cooperates with the SNAREs in bringing the vesicle and plasma membranes together to mediate membrane fusion (*Araç et al., 2006*; *Seven et al., 2013*). Induction of membrane curvature by insertion of the $C_2$ domain $Ca^{2+}$-binding loops into the bilayer was also proposed to stimulate membrane fusion (*Martens et al., 2007*). While these ideas are attractive, the mechanism of action of Syt1 remains enigmatic, in part because dozens of papers have described Syt1-SNARE interactions but it is unclear which of these interactions is physiologically relevant and which ones arise merely from the high promiscuity of these proteins (*Jahn and Scheller, 2006*; *Rizo et al., 2006*). Moreover, Syt1 is believed to function in a tight interplay with complexins whereby complexins play both inhibitory and stimulatory functions and $Ca^{2+}$ binding to Syt1 releases the inhibition (*Giraudo et al., 2006*; *Schaub et al., 2006*; *Tang et al., 2006*), but the underlying mechanism is unknown.

Potentially critical insights were provided by three structures of Syt1-SNARE complexes (*Brewer et al., 2015*; *Zhou et al., 2015*; *Zhou et al., 2017*), but the observed binding modes were drastically different (*Figure 1A–G*). Analysis in solution by NMR spectroscopy revealed a dynamic interaction involving the polybasic region of the Syt1 $C_2B$ domain, while R398,R399 at the bottom of $C_2B$ remain available for membrane binding (*Brewer et al., 2015*; *Figure 1A*). The functional relevance of this binding mode was supported by the finding that release was disrupted by mutations of basic residues that contact the SNAREs (including R322E,K325E), but not by nearby basic residues

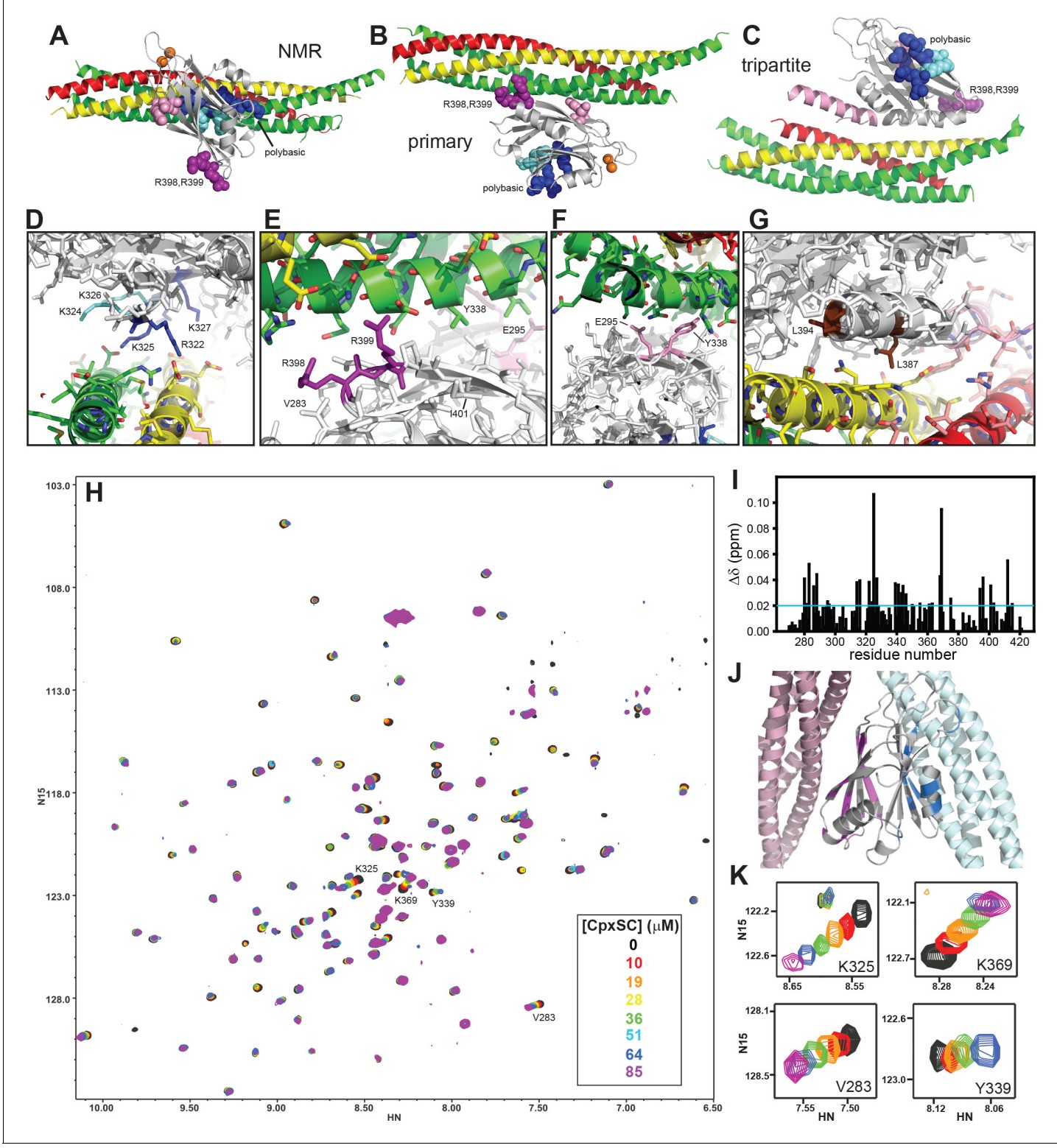

**Figure 1.** The Syt1 $C_2B$ domain binds to CpxSC through the polybasic region and the primary interface in solution. (A–C) Ribbon diagrams illustrating the structures of the Syt1-SNARE complex solved by NMR spectroscopy (A) or X-ray crystallography without (B) or with a bound complexin-1 fragment (C), which revealed the binding modes mediated by the polybasic region, the primary interface and the tripartite interface, respectively. The PDB accession codes for the structures are 2N1T, 5KJ7 and 5W5C, respectively. Syntaxin-1 is in yellow, synaptobrevin in red, SNAP-25 in green, complexin-1 in pink and the Syt1 $C_2B$ domain in gray, with bound $Ca^{2+}$ ions shown as orange spheres. The side chains of several residues from the polybasic region are shown as dark blue (K313, K321, R322, K325 and K327) or cyan spheres (K324 and K326), those of R398,R399 in primary region II as purple spheres

*Figure 1 continued on next page*

Figure 1 continued

and those of E295 and Y338 in primary region I as pink spheres. (D–G) Close-up views of the polybasic (D), primary region II (E), primary region I (F) and tripartite (G) interfaces. All side chains are shown as stick models, with the same color used in (A–C) for the $C_2B$ domain. For the SNARE complex, nitrogen atoms are in blue, oxygen atoms in red and carbon atoms in green (SNAP-25), yellow (syntaxin-1) and pink (complexin-1 and synaptobrevin). The L387 and L394 residues are colored in brown in panel G. Selected side chains are labeled. Note that the structure determined by NMR spectroscopy was highly dynamic and panels (A,D) show just one member of this dynamic ensemble that illustrates the common key feature of the ensemble, namely the involvement of the residues colored in dark blue in binding to a polyacidic patch of the SNARE complex. (H) Superposition of $^1H$-$^{15}N$ TROSY-HSQC spectra of $^2H$,$^{15}N$-IM-$^{13}CH_3$-$C_2B$ domain in the absence of $Ca^{2+}$ and the presence of different concentrations of CpxSC as indicated by the color code ($C_2B$ concentrations gradually decreased from 32 to 12 μM). Cross-peaks broadened gradually with increasing CpxSC concentrations and contour levels were adjusted to allow observation of most cross-peaks in each spectrum. Some cross-peaks broadened beyond detection at high CpxSC concentrations. (I) Chemical shift changes induced by 51 μM CpxSc on the $^1H$-$^{15}N$ TROSY-HSQC cross-peaks of the $C_2B$ domain. Composite $\Delta\delta$ values were calculated as $[(\Delta\delta^1H)^2+(0.17*\Delta\delta^{15}N)^2]^{1/2}$, where $\Delta\delta^1H$ is the chemical shift change in the $^1H$ dimension and $\Delta\delta^{15}N$ is the chemical shift change in the $^{15}N$ dimension. (J) Summary of the largest chemical shift changes caused by 51 μM CpxSc on the $^1H$-$^{15}N$ TROSY-HSQC cross-peaks of the $C_2B$ domain. A ribbon diagram of the $C_2B$ domain (gray) with a SNARE complex bound to the primary interface on the left (light pink) and SNARE complex bound to the polybasic region on the right (light blue) is shown. $C_2B$ domain residues corresponding to cross-peaks with $\Delta\delta > 0.02$ ppm (blue line in panel I) are highlighted in blue (polybasic region) or pink (primary interface). (K) Expansions showing the changes observed at increasing CpxSC concentrations in the cross-peaks corresponding to K325 and K369 at the polybasic region, and V283 and Y339 at the primary interface. Only spectra at selected concentrations of CpxSC are shown. The color code is the same as in panel H.

The online version of this article includes the following figure supplement(s) for figure 1:

**Figure supplement 1.** Cpx1(26-83) prevents aggregation of Syt1 $C_2B$-SNARE complexes.

**Figure supplement 2.** Superposition of $^1H$-$^{15}N$ TROSY-HSQC spectra of $^2H$,$^{15}N$-IM-$^{13}CH_3$-$C_2B$ domain in the presence of 1 mM $Ca^{2+}$ and the presence of different concentrations of CpxSC as indicated by the color code ($C_2B$ concentrations gradually decreased from 32 to 12 μM).

**Figure supplement 3.** Assignments of the $^1H$-$^{15}N$ TROSY-HSQC cross-peaks of the Syt1 $C_2B$ domain in 20 mM HEPES (pH 7.4), 100 mM KCl, 1 mM EDTA, 1 mM TCEP.

that do not point toward the SNAREs (K324E,K326E) (*Figure 1D*), while both mutations disrupted $PIP_2$ binding similarly (*Brewer et al., 2015*). The second structure, determined by X-ray crystallography, revealed a so-called primary interface involving two regions of the $C_2B$ domain, one formed largely by E295 and Y338 (region I) and another including R398,R399 (region II) (*Figure 1B,E, F*; *Zhou et al., 2015*). In this structure, the polybasic region is available for $PIP_2$ binding, an arrangement that was supported by a low-resolution cryo-electron microscopy (cryo-EM) structure of Syt1-SNARE complexes coating lipid tubes (*Grushin et al., 2019*). The physiological relevance of the primary interface was supported by the strong impairments of release caused by mutations in both region I and II (E295A/Y338W and R398Q/R399Q). However, Syt1-SNARE co-immunoprecipitation (co-IP) was not significantly altered by the R398Q/R399Q mutation and only moderately disrupted by the E295A/Y338W mutation (*Zhou et al., 2015*).

X-ray crystallography of a tripartite Syt1-complexin-1-SNARE complex yielded a third structure (*Zhou et al., 2017*) where complexin-1 forms an α-helix that binds to a groove between synaptobrevin and syntaxin-1, as observed for a binary complexin-1-SNARE complex (*Chen et al., 2002*), and an α-helix of the $C_2B$ domain binds to the same groove, continuing the complexin-1 helix (*Figure 1C,G*). Isothermal titration calorimetry (ITC) supported the existence of this interaction in solution and suggested that Syt1 cooperates with complexin-1 for binding to the SNARE complex, providing an explanation for the finding that the dominant negative effect on release caused by mutation of the Syt1 $C_2B$ domain $Ca^{2+}$ binding sites (*Mackler et al., 2002*; *Wu et al., 2017*) requires complexin-1 (*Zhou et al., 2017*). The functional importance of the tripartite interface was supported by the observation that a L387Q/L394Q mutation disrupted binding and impaired neurotransmitter release. However, as pointed out in *Zhou et al., 2017*, the primary interface was also observed in these crystals, which had a 1:1:1 Syt1-SNARE complex-complexin-1 stoichiometry. Hence, one of the two interfaces might constitute a crystal contact. Moreover, a screen for mutations that abrogate the dominant negative effect of mutations in the Syt1 $C_2B$ domain $Ca^{2+}$-binding sites in *Drosophila* yielded a large number of mutations in the primary interface but none in the tripartite interface (*Guan et al., 2017*), indicating that the dominant negative effect may require Syt1-SNARE complex binding through the primary rather than the tripartite interface.

The role of $Ca^{2+}$ in Syt1-SNARE interactions has also been enigmatic. Many studies reported that $Ca^{2+}$ strongly stimulates binding of Syt1 to SNAREs and SNARE complexes both in solution and on membranes [e.g. (*Chapman et al., 1995*; *Dai et al., 2007*; *Davis et al., 1999*; *Gerona et al., 2000*;

*Huang and Cafiso, 2008*; *Li et al., 1995*; *Lynch et al., 2007*; *Zhou et al., 2013*)]. Paradoxically, the Syt1 $Ca^{2+}$-binding loops are not involved in SNARE complex binding in the structures described above, although $Ca^{2+}$ is believed to stimulate binding by increasing the positive electrostatic potential of the $C_2B$ domain, as the SNARE complex is negatively charged. Note also that $Ca^{2+}$ disrupted Syt1-SNARE complexes assembled on lipid tubes (*Grushin et al., 2019*), but another study reported binding of Syt1 to membrane-anchored SNARE complex in the absence and presence of $Ca^{2+}$ (*Wang et al., 2016*). Conversely, ATP was reported to abolish Syt1-SNARE complex at physiological ionic strength, with or without $Ca^{2+}$ (*Park et al., 2015*).

To shed light into this strikingly confusing picture and decipher how the functions of Syt1 and the SNAREs are coupled, we performed a systematic analysis of Syt1-SNARE complex interactions in solution and on membranes. NMR data show that a soluble complexin-1-SNARE complex binds to the polybasic region and the primary interface of the Syt1 $C_2B$ domain with similar affinities, but binding to the tripartite interface could not be detected. Analyses of binding of a Syt1 fragment spanning the two $C_2$ domains ($C_2AB$) to nanodiscs or nanodiscs containing anchored SNARE complexes using FRET show that $C_2AB$ binds to membrane-anchored SNARE complexes under a variety of conditions, and that complexin-1 does not affect the affinity. Binding of the SNARE complex to the primary interface is impaired by the R398Q/R399Q mutation but is enhanced by the E295A/Y338W mutation, both in solution and on nanodiscs, showing that the primary interface is involved in binding to the nanodisc-anchored SNARE complex. Importantly, $Ca^{2+}$-dependent binding of $C_2AB$ to SNARE complexes anchored on $PIP_2$-nanodiscs is almost abolished in the presence of ATP, but $Ca^{2+}$-independent binding remains. We also observed that the R322E/K325E mutation disrupts $Ca^{2+}$-dependent binding of $C_2AB$ to $PIP_2$-containing nanodiscs much more strongly than the K324E/K326E mutation, in correlation with the effects of these mutations on release (*Brewer et al., 2015*). Together with previous data, these results suggest a model whereby Syt1 binds to the SNARE complex through the primary interface and to $PIP_2$ through the polybasic region before $Ca^{2+}$ influx; $Ca^{2+}$ binding to the $C_2B$ domain induces a specific, $PIP_2$-dependent interaction with the plasma membrane, disrupting the interaction with the SNARE complex and allowing cooperation between the SNAREs and Syt1 in inducing membrane fusion.

## Results

### The SNARE complex binds to the $C_2B$ domain polybasic region and to the primary interface with similar affinity in solution

The tendency of the Syt1 $C_2$ domains to precipitate with the SNARE complex formed by the SNARE motifs of synaptobrevin, syntaxin-1 and SNAP-25, particularly in the presence of $Ca^{2+}$, hindered analysis of their interactions in solution by NMR spectroscopy (*Dai et al., 2007*; *Zhou et al., 2013*). Inclusion of 125 mM KSCN allowed NMR analyses in the presence of $Ca^{2+}$ (*Brewer et al., 2015*), but KSCN might have favored binding of the SNARE complex to the $C_2B$ domain polybasic region over binding to the primary interface. In attempts to solve the solubility problem without addition of KSCN, we found that including the complexin-1 fragment that we used for crystallization with the SNARE complex [residues 26–83; Cpx1(26-83)] (*Chen et al., 2002*) dramatically improved the solubility of mixtures containing the $Ca^{2+}$-bound $C_2B$ domain and the SNARE complex (*Figure 1—figure supplement 1*). Taking advantage of this observation, we analyzed the binding sites of the SNARE complex on the Syt1 $C_2B$ domain in the presence of Cpx1(26-83) using NMR spectroscopy. Note that the Syt1 $C_2$ domains do not bind to complexin-1 under the conditions that we used for the NMR experiments and that Cpx1(26-83) binds to the SNARE complex with high affinity ($K_D$ ca. 25 nM) (*Xu et al., 2013*). Hence, the assembly formed by Cpx1(26-83) and the SNARE motifs of synaptobrevin, syntaxin-1 and SNAP-25 can be considered as a single complex that we refer to as CpxSC. For improved sensitivity, we prepared samples of the $C_2B$ domain that were $^2H,^{15}N$-labeled and specifically $^{13}CH_3$-labeled at the Ile δ1 and Met methyl groups ($^2H,^{15}N$-IM-$^{13}CH_3$-$C_2B$). First we analyzed the perturbations caused by addition of increasing amounts of CpxSC on transverse relaxation optimized spectroscopy (TROSY)-enhanced $^1H$-$^{15}N$ heteronuclear single quantum coherence (HSQC) spectra of $^2H,^{15}N$-IM-$^{13}CH_3$-$C_2B$. Titrations of $^2H,^{15}N$-IM-$^{13}CH_3$-$C_2B$ with CpxSC were performed in the presence of $Ca^{2+}$ at physiological ionic strength and in the absence of $Ca^{2+}$ at lower salt concentration (100 mM KCl) to enhance binding, as Syt1-SNARE complex binding is weaker in the absence

than in the presence of $Ca^{2+}$ (*Zhou et al., 2013*). Similar cross-peak shifts were observed in both sets of experiments (*Figure 1H*, *Figure 1—figure supplement 2*), indicating that $Ca^{2+}$ does not affect the binding mode(s).

To analyze the data, we obtained assignments of the $^1H$-$^{15}N$ TROSY-HSQC spectrum of $Ca^{2+}$-free $^2H$,$^{15}N$-IM-$^{13}CH_3$-$C_2B$ (*Figure 1—figure supplement 3*) based on the assignments obtained previously in a different buffer (*Fernandez et al., 2001*) and a titration where the buffer composition was gradually changed from one buffer to the other. Mapping of the residues corresponding to the cross-peaks that exhibited the largest shifts induced by CpxSC (*Figure 1I*) onto the structure of the $C_2B$ domain clearly showed that the residues are clustered on two sides of the β-sandwich, one containing the polybasic region and another corresponding to the primary interface (*Figure 1J*). There were no significant shifts in cross-peaks from residues in the tripartite interface, which is largely formed by the α-helix spanning residues 384–395, except for residues at the very C-terminus of this helix that are near the primary interface. We could not obtain accurate $K_{DS}$ for the interactions of CpxSC with the polybasic region and the primary interface from these data because we did not achieved saturation at the highest concentration of CpxSC that we reached (85 μM) and, at these concentrations, some cross-peaks broadened beyond detection or exhibited odd behavior that did not correlate with the previous titration points (illustrated in *Figure 1K* by the cross-peaks from two residues corresponding to the polybasic region, K325 and K369, and two residues from the primary interface, V283 and Y339). However, it was clear that the affinities of both binding sites are comparable, and the $K_{DS}$ are larger than 20 μM.

## Mutation of R322,K325 at the polybasic region and R398,R399 at the primary interface abolishes $C_2B$-SNARE complex binding in solution

We next examined the effects of mutations in the two binding sites of the $C_2B$ domain for the SNARE complex, using $^2H$,$^{15}N$-IM-$^{13}CH_3$-labeled samples of $C_2B$ domain mutants. Since the $C_2B$ domain $Ca^{2+}$-binding region is not involved in binding and $Ca^{2+}$ did not affect the binding modes, these experiments were performed in the absence of $Ca^{2+}$. We first analyzed the effects of the R322E/K325E mutation that we designed previously to disrupt SNARE complex binding to the polybasic region and strongly impaired neurotransmitter release (*Brewer et al., 2015*). A titration with CpxSC revealed substantial shifts for selected cross-peaks of the $C_2B$ mutant HSQC spectrum, but the patterns were somewhat distinct from those observed for the WT $C_2B$ domain (compare *Figure 2A* with 1H). The cross-peaks from the β-strand containing the mutated residues shifted and their positions could not be ascertained. However, we could identify the diagnostic cross-peak from K369, which is in the same face of the $C_2B$ domain as the polybasic region. The position of this cross-peak was unaffected even at 85 μM CpxSC (*Figure 2C*), confirming that binding to the polybasic region was abolished by the R322E/K325E mutation. We also observed that the diagnostic cross-peaks from the primary interface corresponding to V283 and Y339 shifted faster in the titration of the R322E/K325E mutant than observed for the WT $C_2B$ domain (*Figure 2C*), which is a natural consequence of the lack of competition with the polybasic region for binding to CpxSC. Interestingly, multiple cross-peaks from the primary interface exhibited much more substantial CpxSC-induced shifts for the mutant than for the WT $C_2B$ domain, including cross-peaks from R398 and I401 (*Figure 2A,D*), residues that are near V283 (*Figure 1E*). These results suggest that binding of CpxSC to the primary interface is more extensive for the R322E/K325E mutant $C_2B$ and that the polybasic region not only competes with the primary interface for binding to CpxSC but also hinders full $C_2B$-SNARE engagement at the primary interface.

R398 and R399 at the bottom of the $C_2B$ domain are in the primary interface (*Figure 1E*), but the R398Q/R399Q mutation that almost abolishes neurotransmitter release (*Xue et al., 2008*) did not significantly impair Syt1-SNARE co-IP (*Zhou et al., 2015*). To investigate the contribution of these tandem arginines to Syt1-SNARE complex binding at the primary interface, we analyzed binding of CpxSC to the R398Q/R399Q mutant $C_2B$ domain. We observed that the CpxSC-induced shifts of the cross-peaks from the primary interface were almost abolished (*Figure 2B* and expansions for the V283 and Y339 cross-peaks in *Figure 2C*), showing that the mutation strongly impairs binding of CpxSC to this interface. As expected, CpxSC still bound to the polybasic region of R398Q/R399Q $C_2B$ (e.g. K369 cross-peak, *Figure 2C*). Since individual R398Q and R399Q mutations also impair release considerably (*Xue et al., 2008*), we also tested binding of CpxSC to R398Q $C_2B$ and R399Q $C_2B$ mutants. Both single mutations impaired CpxSC binding substantially (*Figure 2—figure*

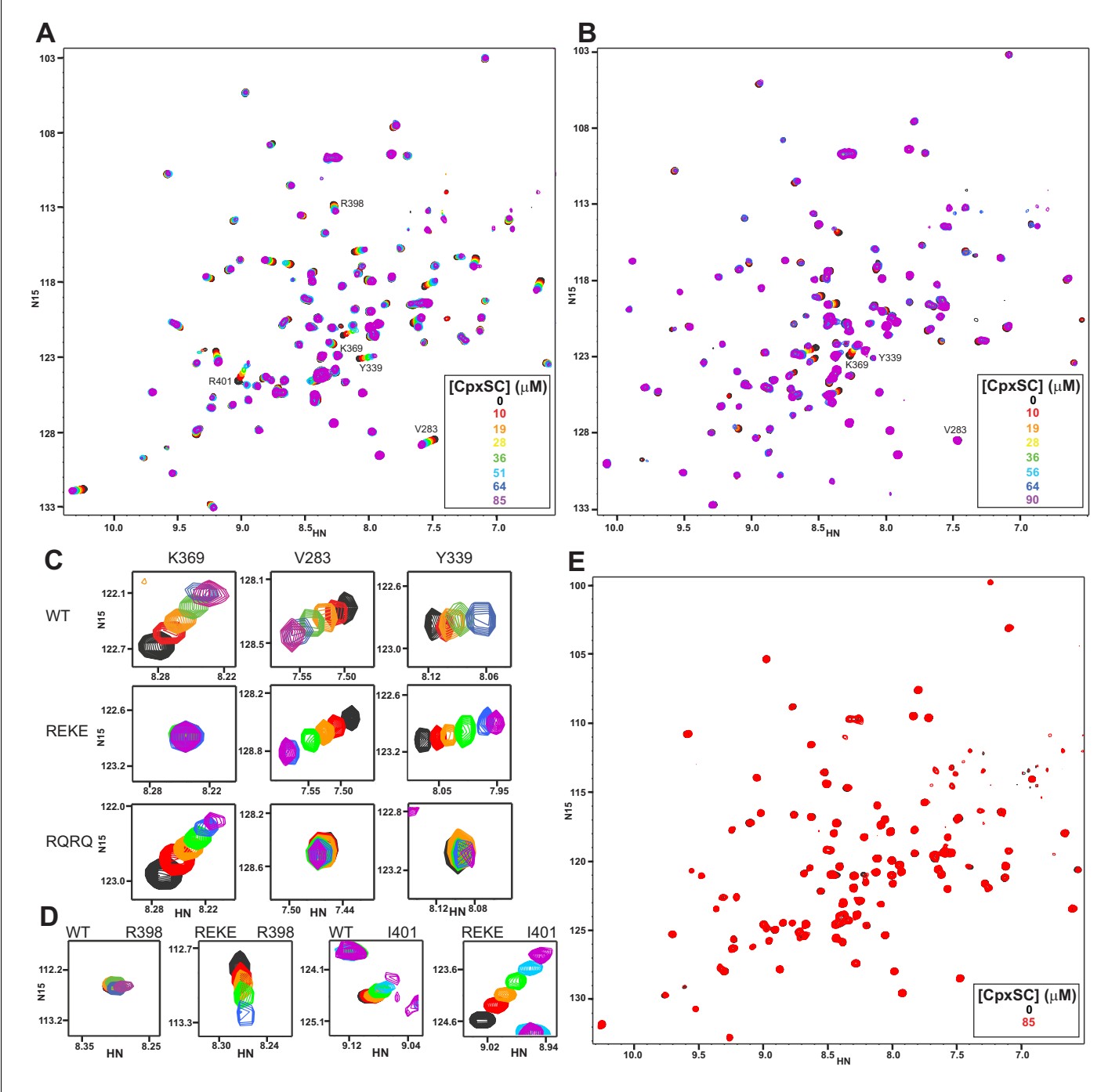

**Figure 2.** Mutations in the polybasic region and the primary interface of the $C_2B$ domain abrogate binding to CpxSC. (A–B) Superposition of $^1H$-$^{15}N$ TROSY-HSQC spectra of R322E/K325E (A) or R398Q/R399Q (B) $^2H$,$^{15}N$-IM-$^{13}CH_3$-$C_2B$ domain in the absence of $Ca^{2+}$ and the presence of different concentrations of CpxSC as indicated by the color code ($C_2B$ concentrations gradually decreased from 32 to 12 μM). Cross-peaks broadened gradually with increasing CpxSC concentrations and contour levels were adjusted to allow observation of most cross-peaks in each spectrum. Some cross-peaks broadened beyond detection at high CpxSC concentrations. (C–D) Expansions showing the changes observed at increasing CpxSC concentrations in selected cross-peaks from WT, R322E/K325E (REKE) or R398Q/R399Q (RQRQ) $C_2B$ domain as indicated by the labels. Only spectra at selected concentrations of CpxSC are shown. The color code is the same as in panels A,B. (E) Superposition of $^1H$-$^{15}N$ TROSY-HSQC spectra of R322E/K325E/R398Q/R399Q $^2H$,$^{15}N$-IM-$^{13}CH_3$-$C_2B$ domain in the absence of $Ca^{2+}$, without (black contours) or with (red contours) 85 μM CpxSC ($C_2B$ concentrations 32 to 12 μM, respectively). Note that a few cross-peaks are missing in the $^1H$-$^{15}N$ TROSY-HSQC spectra of panels A,E (e.g. those of I276 and S279 at the left side of the spectrum) because the deuterated amide groups inside the domain did not exchange to hydrogen in aqueous buffer, likely because the mutations increased the stability of this highly basic domain.

*Figure 2 continued on next page*

*Figure 2 continued*

The online version of this article includes the following figure supplement(s) for figure 2:

**Figure supplement 1.** The individual R398Q and R399Q mutations impair binding of the Syt1 $C_2B$ domain to CpxSC.

**Figure supplement 2.** The R322E/K325E/R398Q/R399Q mutation abrogates binding of the Syt1 $C_2B$ domain to CpxSC.

*supplement 1*), although to a smaller extent than the double R398Q/R399Q mutation. These results show that both R398 and R399 play important roles in binding of Syt1 to CpxSC through the primary interface.

We also prepared a mutant $C_2B$ domain bearing both the R322E/K325E mutation in the polybasic region and the R398Q/R399Q mutation in the primary interface. Titration with CpxSC did not induce significant shifts in the $^1H$-$^{15}N$ TROSY-HSQC spectrum of the R322E/K325E/R398Q/R399Q $C_2B$ mutant even when CpxSC was added at 85 µM concentration (*Figure 2E*). Binding of an unlabeled protein or complex to a $^{15}N$-labeled protein is expected to cause not only cross-peak shifts but also decreased cross-peak intensities due to the larger size of the resulting complex compared to the iso-lated $^{15}N$-labeled protein (*Rizo et al., 2012*), as illustrated in *Figure 2—figure supplement 2A–C* by the gradual decreases in intensities of selected cross-peaks of WT $C_2B$ caused by increasing CpxSC concentrations. In contrast, the cross-peak intensities of R322E/K325E/R398Q/R399Q $C_2B$ did not decrease appreciably as increasing concentrations of CpxSC were added. Thus, the ratios of the intensities in the presence of 85 µM CpxSC versus the intensities in the absence of CpxSC for the same selected cross-peaks of R322E/K325E/R398Q/R399Q $C_2B$ were close to 1 (*Figure 2—figure supplement 2A–C*). Analysis of these ratios for all cross-peaks showed a relatively homogeneous dis-tribution, with an average ratio of 0.947 and some natural variability due to the noise in the data, particularly for the weakest cross-peaks (*Figure 2—figure supplement 2D*). The average ratio for five cross-peaks that are in well-resolved regions of the spectrum and correspond to residues in the α-helix involved in the tripartite interface (T383, G384, L387, R388 and S391) is 0.945. These data show that the quadruple R322E/K325E/R398Q/R399Q mutation abolishes binding of the Syt1 $C_2B$ domain to CpxSC and the mutant does not bind through the tripartite interface under these conditions.

To corroborate this conclusion, we also analyzed samples where Cpx1(26-83) was $^2H$,$^{15}N$-labeled and formed $^2H$,$^{15}N$-CpxSC to analyze perturbations on Cpx1(26-83) upon $C_2B$ domain binding. Addition of WT Syt1 $C_2B$ domain caused substantial broadening in the $^1H$-$^{15}N$ TROSY-HSQC spec-trum of CpxSC, as manifested by decreased intensities in the cross-peaks corresponding to struc-tured parts of Cpx1(26-83) in the complex upon binding to $C_2B$ (*Figure 3A*). Importantly, $C_2B$ did not induce significant shifts in any of the cross-peaks from residues at or near the tripartite interface (residues 66–75; see *Figure 3A*). We note that K73, K74 and K75 are flexible in the complex but their cross-peaks are in unique positions that are very sensitive to changes in their environment (*Chen et al., 2002*; *Pabst et al., 2000*; *Trimbuch et al., 2014*). Hence, the absence of shifts in these cross-peaks and those of residues 66–75 in general show that the $C_2B$ domain does not bind to CpxSC through the tripartite interface under these conditions. Moreover, R322E/K325E/R398Q/ R399Q $C_2B$ did not cause shifts or intensity decreases in the $^1H$-$^{15}N$ TROSY-HSQC spectrum of $^2H$,$^{15}N$-CpxSC (*Figure 3B*), showing again that the quadruple mutation abrogates binding of the Syt1 $C_2B$ domain to CpxSC. Hence, we do not find any evidence for binding of the $C_2B$ domain to CpxSC through the tripartite interface even after abolishing the interactions involving the polybasic region and the primary interface. These results contrast with ITC data suggesting the existence of a Syt1-complexin-1-SNARE complex interaction involving the tripartite interface in experiments per-formed with Syt1 $C_2B$ bearing seven mutations designed to disrupt binding via the polybasic region and the primary interface (KA-Q mutant), and a complexin-1-SNARE complex bearing five mutations to further disrupt such binding (*Zhou et al., 2017*). We are collaborating with the laboratory of Axel Brunger to determine the reasons underlying these conflicting results, and the results of these efforts will be published elsewhere.

Since the L387Q/L394Q mutation in the α-helix of the $C_2B$ domain that forms the tripartite inter-face was reported to disrupt binding through this interfaces based on ITC data (*Zhou et al., 2017*), we tried to analyze whether this mutation perturbs binding of $C_2B$ to CpxSC. The $^1H$-$^{15}N$ TROSY-HSQC spectrum of $^2H$,$^{15}N$-IM-$^{13}CH_3$-labeled L387Q/L394Q $C_2B$ exhibited well-dispersed cross-

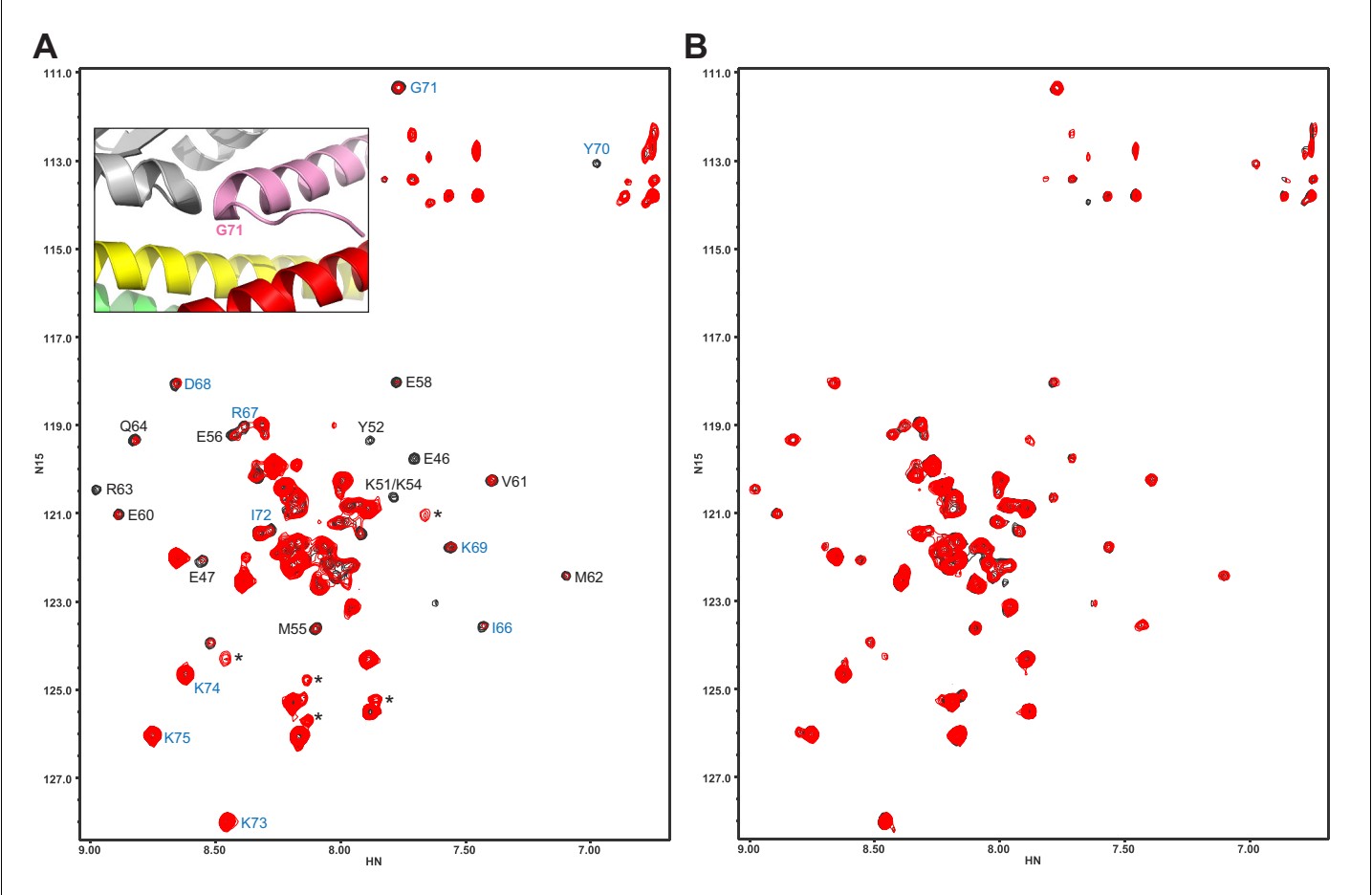

**Figure 3.** The Syt1 $C_2B$ domain does not bind at the tripartite interface of CpxSC at concentrations of tens of micromolar. (**A,B**) $^1H$-$^{15}N$ TROSY HSQC spectra of 40 μM $^2H$,$^{15}N$-Cpx1(26-83) in the absence (black contours) and presence (red contours) of 40 μM WT (**A**) or R322E/K325E/R398Q/R399Q mutant (**B**) $C_2B$ domain. In panel **A**, Assignments of the cross-peaks that are most broadened upon binding to WT $C_2B$ domain are labeled. They correspond to the central α-helix that binds to the SNARE complex and nearby residues. Cross-peaks from other residues do not exhibit as much broadening because they remain flexible. The assignments of residues of Cpx1(26-83) around the corner that contacts the Syt1 $C_2B$ domain in the structure of the tripartite complex are colored in blue. The inset shows a close up of the region where complexin-1 (pink), the $C_2B$ domain (gray) and the SNARE complex (syntaxin-1 yellow, SNAP-25 green, synaptobrevin red) form the tripartite interface. The location of Gly71 is labeled.

The online version of this article includes the following figure supplement(s) for figure 3:

**Figure supplement 1.** The L387Q/L394Q Syt1 $C_2B$ domain mutant is unstable.

peaks characteristic of a folded protein domain, but also contained broad cross-peaks in the center of the spectrum that are commonly observed in unstable proteins that unfold and/or aggregate (*Rizo et al., 2012*; *Figure 3—figure supplement 1*). Addition of 20 μM CpxSC led to strong decreases in cross-peak intensities that arose at least in part from sample precipitation. Hence it was not possible to continue the titration. These results show that the L387Q/L394Q causes a considerable destabilization of the Syt1 $C_2B$ domain, which is consistent with the finding that this mutation decreases the thermal denaturation temperature of the $C_2B$ domain by about 10°C (*Zhou et al., 2017*). Such destabilization may arise because this mutation involves the replacement of a hydrophobic side chain that packs inside the $C_2B$ domain and barely contacts the SNAREs (L394; *Figure 1G*) with a polar residue. Hence, it is plausible that the disruption of neurotransmitter release caused by the L387Q/L394Q mutation (*Zhou et al., 2017*) arose from a general loss of function caused by protein instability.

In summary, our data show that binding of the Syt1 $C_2B$ domain to the Cpx1(26-83)-SNARE complex in solution involves the polybasic region and the primary interface, while binding via the tripartite interface is undetectable under the conditions of our NMR experiments even when CpxSC is

added at 85 µM concentration and binding through the polybasic region and the tripartite interface is abolished. These results indicate that the tripartite interface observed in the complexin-1-Syt1-SNARE complex crystals might have arisen from crystal packing, but further research will be required to clarify this issue.

## The E295A/Y338W mutation in the primary interface enhances $C_2B$-CpxSC binding in solution

The primary interface involves two regions of the $C_2B$ domain, one containing R398 and R399 (region II), and the other E295 and Y338 (region I). The functional importance of region I was demonstrated by the strong disruption of neurotransmitter release caused by the E295A/Y338W mutation in this region (*Zhou et al., 2015*). To examine the effects of this mutation on Syt1-SNARE binding, we performed titrations of $^2H,^{15}N$-IM-$^{13}CH_3$-E295A/Y338W $C_2B$ with CpxSC. Interestingly, we found that CpxSC caused more extensive changes in the $^1H$-$^{15}N$ TROSY-HSQC spectrum of this mutant (*Figure 4A*) than those observed for WT $C_2B$ domain (*Figure 1H*). Analysis of these changes showed that cross-peaks from residues in or near the polybasic region shifted less extensively and more slowly than observed for WT $C_2B$, particularly in the initial points of the titration (*Figure 4B,C*). Conversely, cross-peaks from the primary interface moved faster and more extensively for E295A/Y338W $C_2B$ than for WT $C_2B$; indeed, some cross-peaks that did not shift or barely shifted in WT $C_2B$ exhibited considerable shifts for E295A/Y338W $C_2B$ domain, including the R398 cross-peak (*Figure 4B,D*). These effects in cross-peaks from the primary interface are reminiscent of those observed for the R322E/K325E mutant $C_2B$ domain (*Figure 2A,C,D*), but the shifts observed for the primary interface of the E295A/Y338W $C_2B$ mutant are even larger. These results show that the E295A/Y338W mutation actually increases the affinity of the Syt1 $C_2B$ domain for CpxSC rather than impairs binding, and also appears to make the interaction at the primary interface more extensive, as observed for the R322E/K325E mutation. The basis for this behavior is unclear, but the increased affinity caused by the E295A/Y338W mutation may arise from replacing a tyrosine with a tryptophan, which increases the hydrophobic surface area of this residue. This change may be readily accommodated because the packing in this region of the interface with the SNARE complex is not optimal. We note that the movement of some cross-peaks with increasing CpxSC concentration was curved in some cases (e.g. those of K325 and R398, *Figure 4C,D*). This finding suggests that there is an interplay between binding of CpxSC to the primary interface and to the polybasic region. CpxSC appears to bind exclusively to one or the other site at low concentrations, likely because of steric hindrance disfavors simultaneous binding of two CpxSC complexes to one $C_2B$ domain. However, simultaneous binding to both sites might be allowed at higher CpxSC concentrations by slight alterations in both binding modes, leading to the curved cross-peak movement.

Because the WT and mutant $C_2B$ domains used for all the titrations with CpxSC were specifically $^{13}CH_3$-labeled at the Met and Ile $\delta1$ methyl groups, we also acquired $^1H$-$^{13}C$ heteronuclear multiple quantum coherence (HMQC) spectra of each sample, as these spectra offers very high sensitivity (*Ruschak and Kay, 2010*). The HMQC spectra contained only a small number of probes (*Figure 4—figure supplement 1A*) and hence provided more limited information than the $^1H$-$^{15}N$ TROSY-HSQC spectra, but corroborated the conclusions obtained from the latter regarding binding to the primary interface. Thus, CpxSC caused shifts in the cross-peak from the I401 $\delta1$ methyl group at the primary interface that were more marked for the R322E/K325E mutant, were even larger for the E295A/Y338W mutant, and were abolished for the R398Q/R399Q and R322E/K325E/R398Q/R399Q mutants (*Figure 4—figure supplement 1B*). The cross-peak from the $\delta1$ methyl of I293, another residue at the primary interface, was shifted by CpxSC only for the E295A/Y338W mutant.

## Syt1 $C_2AB$ binds simultaneously to membranes and the SNARE complex

Our NMR data show that there are two major binding modes between the Syt1 $C_2B$ domain and the SNARE complex in solution, one involving the polybasic region of $C_2B$ and the other involving the primary interface, which includes the tandem arginines R398,R399. Since the polybasic region and the tandem arginines have been implicated also in membrane binding [e.g. (*Araç et al., 2006*; *Bai et al., 2004*; *Li et al., 2006*; *Xue et al., 2008*)], it is critical to analyze interactions between Syt1 and membrane-anchored SNARE complexes to assess whether either of these two binding modes still remain in the presence of membranes, or a different type of

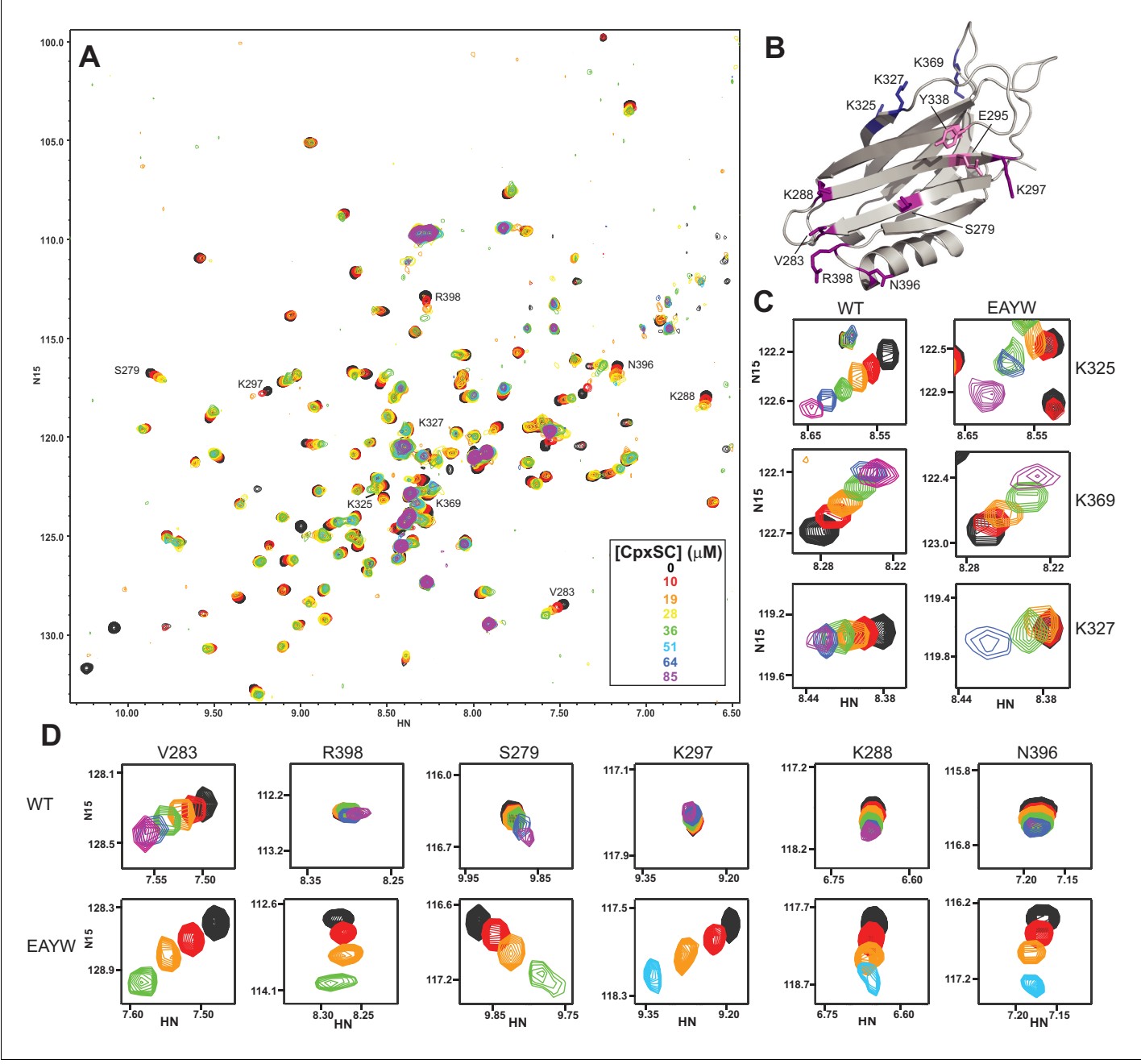

**Figure 4.** The E295A/Y338W mutation enhances the affinity of the Syt1 C$_2$B domain primary interface for CpxSC. (**A**) Superposition of $^1$H-$^{15}$N TROSY-HSQC spectra of E295A/Y338W $^2$H,$^{15}$N-IM-$^{13}$CH$_3$-C$_2$B domain in the absence of Ca$^{2+}$ and the presence of different concentrations of CpxSC as indicated by the color code (C$_2$B concentrations gradually decreased from 32 to 12 μM). Cross-peaks broadened gradually with increasing CpxSC concentrations and contour levels were adjusted to allow observation of most cross-peaks in each spectrum. Some cross-peaks broadened beyond detection at high CpxSC concentrations. (**B**) Ribbon diagram of the Syt1 C$_2$B domain (PDB accession code 1UOV) showing the location of the residues corresponding to the cross-peaks highlighted in the other panels to illustrate that the enhanced binding caused by the E295A/Y338W mutations occurs at the primary interface. (**C–D**) Expansions showing the changes observed at increasing CpxSC concentrations in selected cross-peaks from WT and E295A/Y338W mutant C$_2$B domain as indicated by the labels. Only spectra at selected concentrations of CpxSC are shown. The color code is the same as in panel **A**.

The online version of this article includes the following figure supplement(s) for figure 4:

**Figure supplement 1.** Titrations of WT and mutant Syt1 C$_2$B domains with CpxSC monitored by $^1$H-$^{13}$C HMQC spectra.

interaction might occur. For this purpose, we designed a strategy based on anchoring SNARE complexes on nanodiscs. For these studies we used the Syt1 $C_2AB$ fragment that spans both $C_2$ domains because the $C_2A$ domain contributes to binding of Syt1 to membranes and could contribute to binding to the SNARE complex directly or indirectly, for example through cooperativity between SNARE and lipid interactions of Syt1. As a scaffold for the nanodiscs, we chose MSP1E3D1 because it yields stable nanodiscs with a diameter of ca. 13 nm (*Nath et al., 2007*) that can accommodate the SNARE complex and allow potential simultaneous interactions of Syt1 $C_2AB$ with the SNARE complex and the lipids. We prepared SNARE complexes with full-length syntaxin-1 and the SNARE motifs of SNAP-25 and synaptobrevin, anchoring the complexes on the nanodiscs through the syntaxin-1 transmembrane (TM) region to mimic the configuration expected to occur on the plasma membrane. The stoichiometry of MSP1D3 to syntaxin-1 was adjusted to form nanodiscs that on average contained one SNARE complex. We refer to these macromolecular assemblies as cisSC-NDs, while control nanodiscs prepared without the SNAREs are abbreviated as NDs (*Figure 5A*). Compared to experiments using liposomes, this modular design facilitates analysis of individual SNARE complex-Syt1-membrane assemblies without complications that might arise from liposome clustering induced by $C_2AB$ (*Araç et al., 2006*).

To monitor binding of $C_2AB$ to NDs or cisSC-NDs by FRET, we generated a single-cysteine $C_2AB$ mutant with the cysteine replacing E346, and labeled it with an Alex488 donor fluorescent probe ($C_2AB^*$). This position was chosen because residue 346 is not located on any of the interfaces that have been implicated in SNARE complex binding (*Figure 5—figure supplement 1A*) and placing the fluorescent probe on this residue is not expected to disrupt these interfaces. Unless otherwise indicated, NDs contained 5% rhodamine-labeled phosphatidylethanolamine (Rho-PE), which constitutes a suitable acceptor probe for highly efficient FRET with Alexa488. Indeed, titration of $C_2AB^*$ with cisSC-NDs formed with a 80:15:5 mixture of phosphatildylcholine (PC), phosphatidylserine (PS) and Rho-PE in the presence of 125 mM KCl and 1 mM $Ca^{2+}$ led to progressively more efficient FRET that maximized at a FRET efficiency of ca. 0.88 (*Figure 5B and C*, blue curve). Control experiments with analogous NDs lacking SNARE complex showed that saturation required higher ND concentrations (*Figure 5C*, red curve), yielding a substantially higher apparent $K_D$ (129.5 nM compared to 26.6 nM for the cisSC-NDs). A summary of the $K_D$s obtained under these and other conditions described below is presented in *Supplementary file 1*. *Supplementary file 2* lists cooperativity factors calculated from $K_D$ NDs/$K_D$ cisSC-NDs, which yield an idea of the synergy between interactions of $C_2AB^*$ with the SNAREs and the lipids, for each condition. *Supplementary file 1* also lists repeat experiments performed under selected conditions with different nanodisc preparations that show the reproducibility of the data (see statistics in Materials and methods). We observed a natural variability in the apparent $K_D$s that may arise in part from different incorporation of SNARE complexes into the nanodiscs. Note also that some of the $K_D$s are in the low nM range and may not be accurate because the concentration of $C_2AB^*$ used for all experiments was 50 nM and the corresponding titration curves thus approach saturating binding conditions. Hence, these $K_D$s and corresponding cooperativity factors must be interpreted with caution. For all these reasons, the conclusions described below were obtained by comparing experiments performed on the same day or over period of two days with the same preparations, and were confirmed by additional comparisons made with other preparations on different days. Importantly, the key conclusions are supported by the overall consistency of the data obtained under different conditions.

The higher affinity of $C_2AB^*$ for cisSC-NDs than for NDs (*Figure 5C*, blue and red curves, respectively) shows that $C_2AB^*$ binds simultaneously to the SNARE complex and the lipids under these conditions. Since Syt1 $C_2AB$ is able to bind simultaneously to two membranes in the presence of $Ca^{2+}$ (*Araç et al., 2006*), which could cooperate with binding to the SNARE complex (*Brewer et al., 2015*), we also investigated binding of $C_2AB$ to trans-SNARE complexes formed between two nanodiscs (transSC-NDs). Because we wanted to compare binding to cis and trans SNARE complexes, in these experiments we placed a tetramethylrhodamine (TMR) acceptor fluorescent probe on the SNARE complex rather than on the nanodiscs to allow direct quantification of the SNARE complex concentrations from the UV-vis absorption of the probe. We chose residue 34 of SNAP-25 to place the acceptor probe because it is predicted to be sufficiently close to the donor probe at residue 346 of $C_2AB$ in any of the three structures of Syt1-SNARE complexes that have been elucidated (*Figure 5—figure supplement 1B*), such that binding would be detected regardless of which binding

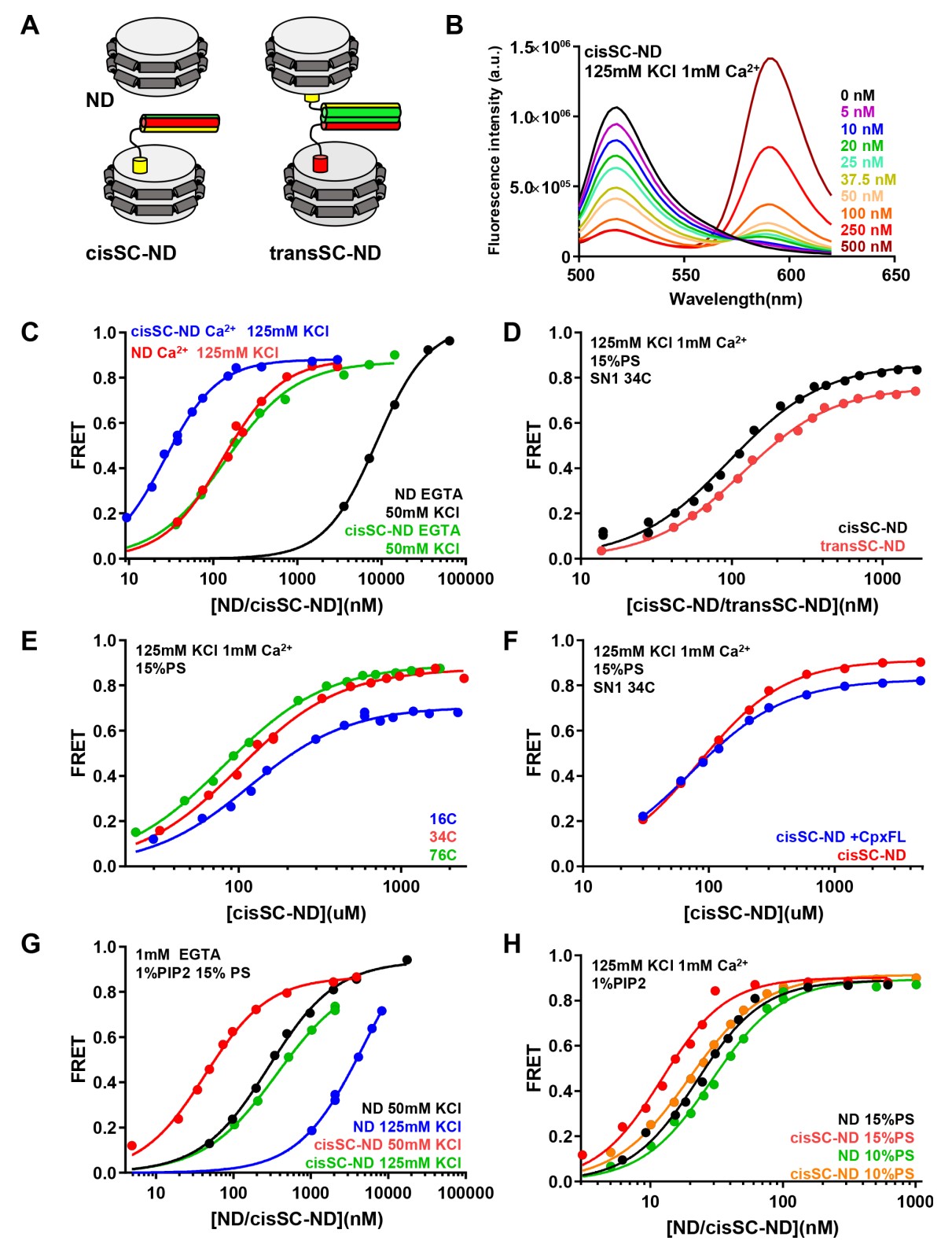

**Figure 5.** Analysis of interactions between Syt1 $C_2AB^*$ and SNARE complexes anchored on nanodiscs using FRET. (**A**) Diagrams illustrating NDs, cisSC-NDs and transSC-NDs. Syntaxin-1 is in yellow, SNAP-25 in green and synaptobrevin in red. Although we used full-length syntaxin-1 to make the nanodisc-anchored SNARE complexes, the N-terminal region preceding the SNARE motif of syntaxin-1 is not shown for simplicity. (**B**) Fluorescence emission spectra of $C_2AB^*$ (labeled with Alexa488 at residue 346) in the presence of increasing concentrations of cisSC-NDs (15% PS, 5% Rho-PE), in

*Figure 5 continued on next page*

Figure 5 continued

125 mM KCl and 1 mM Ca$^{2+}$. (**C**) FRET efficiencies observed in titrations of C$_2$AB* with NDs or cisSC-NDs (15% PS, 5% Rho-PE) in the presence of 50 mM KCl and 1 mM EGTA, or 125 mM KCl and 1 mM Ca$^{2+}$. (**D**) FRET efficiencies observed in titrations of C$_2$AB* with cisSC-NDs or transSC-NDs (15% PS, 5% Rho-PE) in the presence of 125 mM KCl and 1 mM Ca$^{2+}$. (**E**) FRET efficiencies observed in titrations of C$_2$AB* with cisSC-NDs (15% PS) labeled with Rho at position 16, 34 or 76 of SNAP-25 in the presence of 125 mM KCl and 1 mM Ca$^{2+}$. (**F**) FRET efficiencies observed in titrations of C$_2$AB* with cisSC-NDs (15% PS) labeled at residue 34 of SNAP-25 in the presence of 125 mM KCl and 1 mM Ca$^{2+}$, with or without complexin-1. (**G**) FRET efficiencies observed in titrations of C$_2$AB* with NDs or cisSC-NDs (15% PS, 5% Rho-PE, 1% PIP$_2$) in the presence of 1 mM EGTA and 50 or 125 mM KCl. (**H**) FRET efficiencies observed in titrations of C$_2$AB* with NDs or cisSC-NDs (15% PS, 5% Rho-PE, 1% PIP$_2$; or 10% PS, 5% Rho-PE, 1% PIP$_2$) in the presence of 125 mM KCl and 1 mM Ca$^{2+}$. All data were fit with a Hill equation (see Materials and methods).

The online version of this article includes the following source data and figure supplement(s) for figure 5:

**Source data 1.** Summary of apparent KDs.
**Figure supplement 1.** Locations of the fluorescent probes used for the titrations of C$_2$AB* with NDs, cisSC-NDs and transSC-NDs.
**Figure supplement 2.** Models illustrating plausible and impossible binding modes of the Syt1 C$_2$B domain to the SNARE complex on nanodiscs.

mode occurs. Titrations of C$_2$AB* with transSC-NDs and cisSC-NDs yielded similar results (*Figure 5D*) and comparable apparent K$_D$s (118 nM and 96 nM, respectively). The lower K$_D$ obtained with cisSC-NDs labeled with 5% Rho-PE can be attributed to the negative charge added by the labeled lipids, which should increase their membrane affinity for C$_2$AB (*Zhang et al., 1998*). These results suggest that the affinity of C$_2$AB* for cisSC-NDs and transSC-NDs is similar and ensuing experiments were performed with cisSC-NDs for simplicity.

We also analyzed binding of C$_2$AB* to cisSC-NDs that were labeled at the N- or C-terminus of the SNARE complex (residue 16 or 76 of SNAP-25; *Figure 5—figure supplement 1B*). We obtained similar binding curves and comparable K$_D$s to those obtained with the label at residue 34 (*Figure 5E*; *Supplementary file 1*). The FRET efficiencies observed at saturating concentrations were similar for the labels at residues 34 and 76 of SNAP-25, and somewhat lower for the label at residue 16. The FRET efficiencies are consistent with the binding modes involving the polybasic region and the primary interface, which predict that the probe on C$_2$AB* is located at comparable, short distances from residues 34 and 76, and farther from residues 16, but are not consistent with the tripartite complex, where residue 346 of C$_2$AB* is expected to be much closer to residue 16 than to residue 76 (*Figure 5—figure supplement 1B*). However, these results are not conclusive, as other binding modes could also be consistent with the observed FRET efficiencies. We also analyzed the effect of including complexin-1 on binding of C$_2$AB* to cisSC-NDs labeled at residue 34 of SNAP-25 and observed similar binding curves and apparent K$_D$s (*Figure 5F*; *Supplementary file 1*), suggesting that complexin-1 does not substantially alter the interaction of C$_2$AB* with the nanodisc-anchored SNARE complex. This observation is also consistent with binding of C$_2$AB to the SNARE complex through the primary and polybasic interfaces, which is not expected to be affected by complexin-1.

In subsequent experiments we focused on comparing binding of C$_2$AB* to NDs and cisSC-NDs labeled with Rho-PE to analyze the increases in affinity caused by the presence of the SNARE complex, and analyzed how the conditions of the experiments affect the underlying affinities. In experiments performed in 125 mM KCl and 1 mM EGTA to analyze Ca$^{2+}$-independent interactions, binding to NDs was very weak. We lowered the KCl concentration to 50 mM to facilitate binding and were able to observe efficient binding at relatively high ND concentrations (*Figure 5C*, black curve), with an apparent K$_D$ of 8.7 µM. Importantly, we observed a much higher affinity for cisSC-NDs (*Figure 5C*, green curve), with an apparent K$_D$ of 141 nM. These results suggest that there is a strong synergy between binding of Ca$^{2+}$-free C$_2$AB* to the SNARE complex and the nanodisc phospholipids. Since PIP$_2$ enhances binding of the C$_2$B domain to membranes due to interactions with the polybasic region (*Bai et al., 2004*; *Li et al., 2006*), we performed titrations with NDs and cisSC NDs containing 1% PIP$_2$ and indeed observed much higher affinities at 50 mM KCl (*Figure 5G*, black and red curves; apparent K$_D$s 286 nM and 43 nM, respectively). Binding was weaker in 125 mM KCl (*Figure 5G*, blue and green curves) but there was still a large difference in the K$_D$s observed for NDs and cisSC-NDs (apparent K$_D$s 4.9 µM and 366 nM, respectively). We also analyzed binding of C$_2$AB* to NDs and cisSC-NDs containing 1% PIP$_2$ in the presence of 1 mM Ca$^{2+}$ and again observed increased affinity for the latter (*Figure 5H*, black and red curves; apparent K$_D$s 22 nM and 12 nM, respectively). Since the high affinity for cisSC-NDs containing PIP$_2$ implied that we were close to

saturation binding conditions, which hindered analysis of the SNARE-induced enhancement on binding, we also performed experiments with NDs and cisSC-NDs containing 10% instead of 15% PS (i.e. composed of PC:PS:Rho-PE:PIP$_2$ 84:10:5:1) and indeed observed somewhat weaker binding (*Figure 5H*, green and orange curves; apparent $K_D$s 29 nM and 20 nM, respectively). Overall, the increases in affinity caused by simultaneous binding of C$_2$AB* to the nanodiscs and the SNARE complex were considerably lower in the presence than in the absence of Ca$^{2+}$, particularly when the nanodiscs contained PIP$_2$ (*Supplementary file 1*, *2*).

## Mutations in the polybasic region and R398,R399 disrupt Ca$^{2+}$-independent binding of C$_2$AB* to nanodisc-anchored SNARE complex

To dissect the contributions of the polybasic region and R398,R399 at the primary interface to binding of C$_2$AB* to NDs and cisSC-NDs, we performed titrations under various conditions using WT and mutant versions of C$_2$AB* that were labeled with a donor probe at residue 346 and bore the R322E/K325E, R398Q/R399Q or R322E/K325E/R398Q/R399Q mutations (*Figure 6*; *Figure 6—figure supplements 1–6*). Since the much stronger effects of the R322E/K325E mutation than the K324E/K326E mutation on SNARE complex binding and on neurotransmitter release supported the physiological relevance of Syt1-SNARE complex binding through the polybasic region (*Brewer et al., 2015*), we also included C$_2$AB* with the K324E/K326E mutation in these analyses. All NDs and cisSC-NDs included 5% Rho-PE. Ca$^{2+}$-independent binding of C$_2$AB* to NDs was markedly impaired by all mutations, but mutations in the polybasic region impaired binding to NDs containing PIP$_2$ much more strongly than the R398Q/R399Q mutation (*Figure 6A,C*). Hence, binding to these nanodiscs is mediated largely by the polybasic region, as expected, but the tandem arginines also participate in binding to some extent. Binding to cisSC-NDs in EGTA was also impaired strongly by mutations in the polybasic region, but in this case the impairment caused by the R398Q/R399Q mutation was almost as strong (*Figure 6B,D*). All mutations strongly decreased the cooperativity factors calculated from $K_D$ NDs/$K_D$ cisSC-NDs, which provide an idea of the contribution of C$_2$AB*-SNARE complex interactions to cisSC-ND binding and ranged from 6.64 to 62.1 for WT C$_2$AB* in the three conditions including EGTA (*Supplementary file 2*). Thus, both the tandem arginines (R398,R399) and the polybasic region are important for Ca$^{2+}$-independent, simultaneous binding of C$_2$AB* to the SNARE complex and the lipids. These findings suggest that the underlying interactions are dynamic and involve at least two binding modes whereby either the primary interface containing R398,R399 or the polybasic region interacts with the SNAREs and the other basic sequence (the polybasic region or R398,R399) binds to the lipids (*Figure 5—figure supplement 2A,B*).

## Ca$^{2+}$- and PIP$_2$-dependent binding of the C$_2$B domain to membranes hinders SNARE complex binding

An interestingly different picture emerged in experiments performed in the presence of Ca$^{2+}$. Importantly, the R322E/K325E mutation impaired Ca$^{2+}$-dependent binding to PIP$_2$-containing NDs much more dramatically than the K324E/K326E mutation (*Figure 6G*, *Figure 6—figure supplement 6A*; apparent $K_D$s 49–62 nM for the K324E/K326E mutant and 1.5–2.9 μM for R322E/K325E). This striking difference correlates with the effects of these mutations on neurotransmitter release (*Brewer et al., 2015*) and was not observed for Ca$^{2+}$-dependent binding of C$_2$AB* to NDs lacking PIP$_2$ (*Figure 6E*), or Ca$^{2+}$-independent binding to PIP$_2$-containing NDs (*Figure 6A,C*) or liposomes (*Brewer et al., 2015*), all of which were similarly disrupted by the R322E/K325E and K324E/K326E mutations. These findings show that there is a specific Ca$^{2+}$-dependent binding mode of Syt1 to PIP$_2$-containing membranes that involves R322,K325 but not K324,K326, and suggest that the specific, strong disruption of neurotransmitter release induced by the R322E/K325E mutation but not the K324E/K326E mutation arises from impairment of Ca$^{2+}$-dependent binding of Syt1 to the (PIP$_2$-containing) plasma membrane rather than to the SNARE complex.

The cooperativity factors calculated from $K_D$ NDs/$K_D$ cisSC-NDs in the presence of Ca$^{2+}$ under various conditions (*Figure 6E–H*, *Figure 6—figure supplements 4–6*) ranged from 1.39 to 3.11 and hence were considerably smaller than those observed in EGTA (*Supplementary file 2*), indicating that interactions of WT C$_2$AB* with the SNARE complex contributed much less to binding to cisSC-NDs in the presence of Ca$^{2+}$ than in its absence. Interestingly, the cooperativity factors observed for WT C$_2$AB* in the presence of PIP$_2$ were particularly small, suggesting that Ca$^{2+}$-dependent

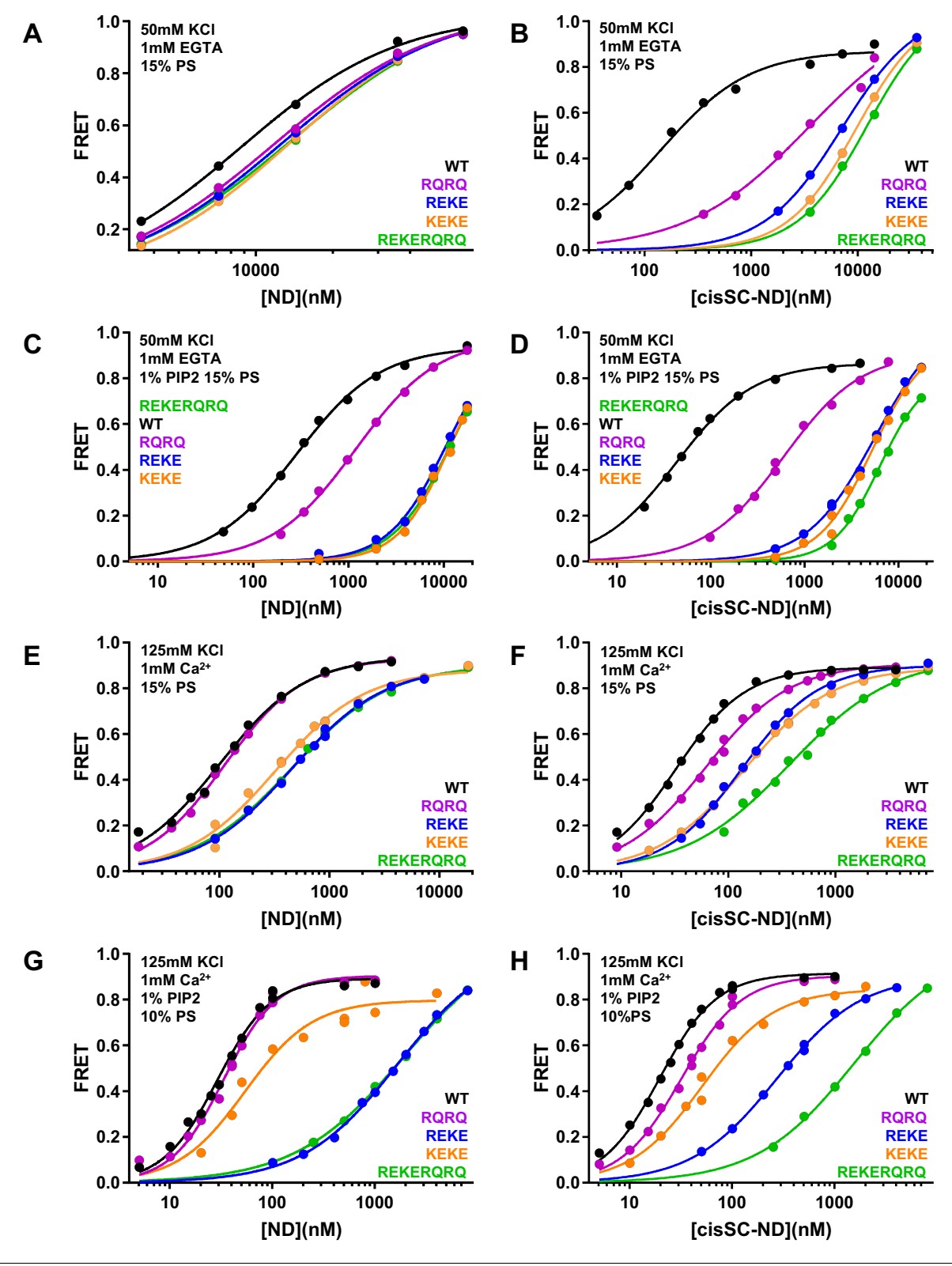

**Figure 6.** Effects of mutations in the Syt1 $C_2B$ domain on $Ca^{2+}$-independent and $Ca^{2+}$-dependent binding of $C_2AB^*$ to NDs and cisSC-NDs. (A–H) FRET efficiencies observed in titrations of WT, R398Q/R399Q (RQRQ), R322E/K325E (REKE), K324E/K326E (KEKE) or R322E/K325E/R398Q/R399Q (REKERQRQ) $C_2AB^*$ with NDs (A,C,E,G) or cisSC-NDs (B,D,F,H) containing 15% PS and 5% Rho-PE (A,B,E,F), 15% PS, 5% Rho-PE and 1% PIP$_2$ (C,D), or

*Figure 6 continued on next page*

Figure 6 continued

10% PS, 5% Rho-PE and 1% PIP$_2$ (**G,H**) in the presence of 1 mM EGTA and 50 mM KCl (**A–D**), or 1 mM Ca$^{2+}$ and 125 mM KCl (**E–H**). All data were fit with a Hill equation (see Materials and methods).

The online version of this article includes the following source data and figure supplement(s) for figure 6:

**Source data 1.** Cooperativity factors.

**Figure supplement 1.** Effects of mutations in the Syt1 C$_2$B domain on Ca$^{2+}$-independent and Ca$^{2+}$-dependent binding of C$_2$AB* to NDs and cisSC-NDs.

**Figure supplement 2.** Effects of mutations in the Syt1 C$_2$B domain on Ca$^{2+}$-independent and Ca$^{2+}$-dependent binding of C$_2$AB* to NDs and cisSC-NDs.

**Figure supplement 3.** Effects of mutations in the Syt1 C$_2$B domain on Ca$^{2+}$-independent and Ca$^{2+}$-dependent binding of C$_2$AB* to NDs and cisSC-NDs.

**Figure supplement 3—source data 1.** Source data for *Figure 6—figure supplement 3*.

**Figure supplement 4.** Effects of mutations in the Syt1 C$_2$B domain on Ca$^{2+}$-independent and Ca$^{2+}$-dependent binding of C$_2$AB* to NDs and cisSC-NDs.

**Figure supplement 5.** Effects of mutations in the Syt1 C$_2$B domain on Ca$^{2+}$-independent and Ca$^{2+}$-dependent binding of C$_2$AB* to NDs and cisSC-NDs.

**Figure supplement 6.** Effects of mutations in the Syt1 C$_2$B domain on Ca$^{2+}$-independent and Ca$^{2+}$-dependent binding of C$_2$AB* to NDs and cisSC-NDs.

**Figure supplement 6—source data 1.** Source data for *Figure 6—figure supplement 6*.

interactions of C2AB* with PIP$_2$ on the nanodiscs preclude interactions with the SNARE complex. The diminished cooperativity factors in the presence of PIP$_2$ were decreased further by the R398Q/R399Q, K324E/K326E and R322E/K325E/R398Q/R399Q mutations, but where substantially increased by the R322E/K325E mutation, presumably because the specific interactions of C$_2$AB* with PIP$_2$ mediated by R322E,K325E were disrupted, allowing binding to the SNARE complex. These findings can be readily rationalized by modeling how the C$_2$B domain binds to a PIP$_2$-containing membrane in a Ca$^{2+}$-dependent manner. Because Ca$^{2+}$ induces binding of the C$_2$B domain in an approximately perpendicular orientation to the membrane that favors insertion of both Ca$^{2+}$-binding loops into the bilayer (*Araç et al., 2006*; *Bai et al., 2004*; *Rufener et al., 2005*), this orientation allows binding of R322 and K325 to PIP$_2$, but K324 and K326 point away from the membrane because they are on the opposite side of the same β-strand (*Figure 5—figure supplement 2C*; see also Figure 10B). In this configuration, interaction of the SNARE complex with the primary interface of the C$_2$B domain is impossible because it would place the syntaxin-1 C-terminus far from the membrane where it is anchored (*Figure 5—figure supplement 2C*). Simultaneous Ca$^{2+}$-dependent binding of the C$_2$B domain to the membrane through its Ca$^{2+}$-binding loops and to the SNARE complex through its polybasic region is in principle compatible with anchoring of syntaxin-1 to the membrane (*Figure 5—figure supplement 2D*), but this binding mode is expected to be prevented by PIP$_2$ because PIP$_2$ binds to K322,R325, the same residues of the polybasic region that are key for SNARE complex binding (*Brewer et al., 2015*).

In summary, Ca$^{2+}$-dependent binding of C$_2$AB* to PIP$_2$-containing membranes is incompatible with the two major Syt1-SNARE complex binding modes, which involve the primary and polybasic interfaces. However, upon Ca$^{2+}$- and PIP$_2$-dependent membrane binding, R398 and R399 can still be involved in non-specific interactions with negative residues of the SNARE complex, and such interactions are also possible if the C$_2$B domain binds in a more slanted orientation to membranes lacking PIP$_2$ (*Figure 5—figure supplement 2E,F*). Such non-specific interactions might underlie the modest decreases in Ca$^{2+}$-dependent binding of C$_2$AB* to cisSC-NDs caused by the R398Q/R399Q mutation (*Figure 6F,H*) and, similarly, the impairments in such binding caused by the K324E/K326E mutation might arise from other non-specific binding modes.

## The E295A/Y338W mutation in the primary interface enhances Ca$^{2+}$-independent binding of C$_2$AB* to cisSNARE complex-nanodiscs

We also analyzed the effects of the E295A/Y338W mutation in region I of the primary interface on binding to NDs and cisSC-NDs. Interestingly, this mutation did not alter binding to NDs under various conditions (*Figure 7*) but caused a considerable increase in affinity for cisSC-NDs containing 15% PS and 1% PIP$_2$ in the absence of Ca$^{2+}$, compared to WT C$_2$AB* (*Figure 7A*; apparent K$_D$s 366

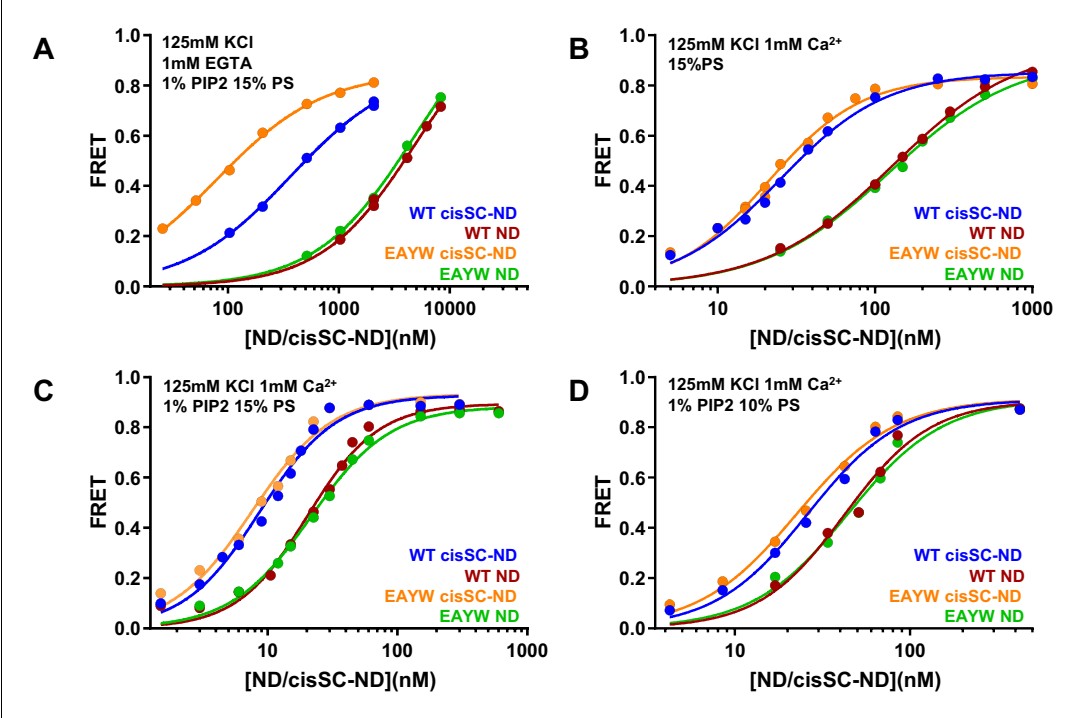

**Figure 7.** The E295A/Y338W mutation in the $C_2B$ domain primary interface enhances binding of $C_2AB^*$ to cisSC-NDs. (A–D) FRET efficiencies observed in titrations of WT or E295A/Y338W (EAYW) $C_2AB^*$ with NDs or cisSC-NDs containing 15% PS, 5% Rho-PE and 1% $PIP_2$ (A,C), 15% PS and 5% Rho-PE (B), or 10% PS, 5% Rho-PE and 1% $PIP_2$ (D) in the presence of 1 mM EGTA and 50 mM KCl (A), or 1 mM $Ca^{2+}$ and 125 mM KCl (B–D). All data were fit with a Hill equation (see Materials and methods).

The online version of this article includes the following source data for figure 7:

**Source data 1.** Source data for *Figure 7*.

nM for WT $C_2AB^*$ and 80 nM for the E295A/Y338W mutant). In contrast, we observed only very slight increases in $Ca^{2+}$-dependent binding of E295A/Y338W $C_2AB^*$ to cisSC-NDs containing 15% PS with or without 1% $PIP_2$, or 10% PS and 1% $PIP_2$, compared to WT $C_2AB^*$ (*Figure 7B–D*). The results obtained in the absence of $Ca^{2+}$ correlate with the increased affinity of the Syt1 $C_2B$ domain for CpxSC observed in our NMR experiments (*Figure 4*) and show that this mutations enhances $Ca^{2+}$-independent binding of $C_2AB^*$ to the SNARE complex within cisSC-NDs via the primary interface. However, the lack of an overt effect of the E295A/Y338W mutation on binding to cisSC-NDs in the presence of $Ca^{2+}$ strongly supports the notion that the primary interface is not involved in $Ca^{2+}$-dependent binding.

## Effects of phospholipids on $C_2AB^*$ binding to the SNARE complex

To gain insights in how the phospholipids in the nanodiscs influence Syt1-SNARE complex interactions, we performed experiments where we monitored direct binding of $C_2AB^*$ to the SNARE complex in cisSC-NDs by placing the FRET acceptor on residue 76 of SNAP-25 (*Figure 8*; *Supplementary file 1*). In these experiments we focused on WT $C_2AB^*$ and $C_2AB^*$ bearing the R398Q/R399Q, R322E/K325E or E295A/Y338W mutations that strongly impair neurotransmitter release. Comparisons of the results obtained with cisSC-NDS and those of parallel experiments performed with soluble SNARE complexes formed with the cytoplasmic region of syntaxin-1 (residues 2–253) (referred to as solubleSC) provided information on how interactions of $C_2AB^*$ with the lipids enhance SNARE complex binding. To examine how negatively charged phospholipid head groups contribute to such enhancement, we performed experiments with cisSC-NDs that contained only PC (cisSC-PC-NDs) or 15% PS,1% $PIP_2$ and 84% PC (cisSC-PC/PS/$PIP_2$-NDs). We also performed experiments with a soluble SNARE complex formed with the syntaxin-1 SNARE motif (residues 191–253) instead of its cytoplasmic region (*Figure 8—figure supplement 1*) to mimic the SNARE complex

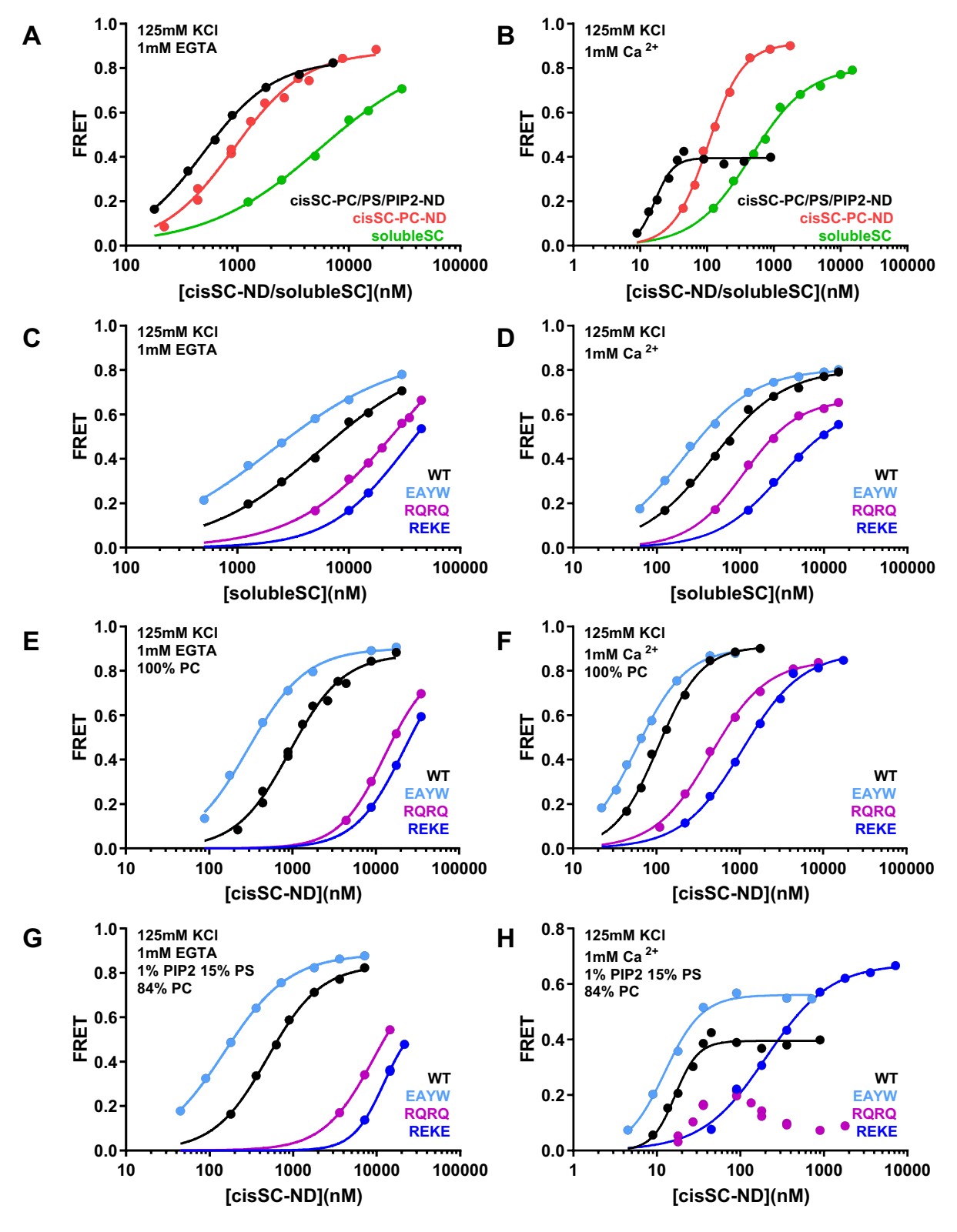

**Figure 8.** Effects of phospholipids on $C_2AB^*$ binding to the SNARE complex. (A–B) FRET efficiencies observed in titrations of WT $C_2AB^*$ with solubleSC, cisSC-PC-NDs (containing only PC) or cisSC-PC/PS/PIP$_2$-NDs (containing 15% PS,1% PIP$_2$, 84% PC) labeled with TMR at residue 76 of SNAP-25 in the presence of 1 mM EGTA (A) or 1 mM $Ca^{2+}$ (B). (C–H) FRET efficiencies in analogous titrations of WT, R398Q/R399Q (RQRQ), R322E/K325E

*Figure 8 continued on next page*

*Figure 8 continued*

(REKE) or E295A/Y338W (EAYW) $C_2AB^*$ with solubleSC (**C–D**), cisSC-PC-NDs (**E–F**) or cisSC-PC/PS/PIP$_2$-NDs (**G–H**) in the presence of 1 mM EGTA (**C, E, G**) or 1 mM Ca$^{2+}$ (**D, F, H**). All data were fit with a Hill equation (see Materials and methods).

The online version of this article includes the following source data and figure supplement(s) for figure 8:

**Source data 1.** Source data for *Figure 8*.
**Figure supplement 1.** Effects of phospholipids on $C_2AB^*$ binding to the SNARE complex.
**Figure supplement 1—source data 1.** Source data for *Figure 8—figure supplement 1*.

used in the NMR experiments (referred to as mcc). The results were similar to those obtained with solubleSC but revealed somewhat weaker affinities for $C_2AB^*$ that are consistent with the NMR data obtained with the $C_2B$ domain and CpxSC (*Figure 1*) and suggest that the N-terminal region of syntaxin-1 contributes to Syt1-SNARE interactions, as proposed previously (*Fernandez et al., 1998*).

In the absence of Ca$^{2+}$, $C_2AB^*$ bound more strongly to cisSC-PC-NDs than to solubleSC ($K_D$ 0.93 and 5.5 µM, respectively), and binding to cisSC-PC/PS/PIP$_2$-NDs was even tighter ($K_D$ 0.49 µM) (*Figure 8A*). Binding of $C_2AB^*$ to these complexes was strongly impaired by the R322E/K325E mutation and to a lesser extent by the R398Q/R399Q mutation, and was strengthened by the E295A/Y338W mutation (*Figure 8C,E,G*). The cooperativity factors calculated from $K_D$ solubleSC/$K_D$ cisSC-PC-NDs and $K_D$ solubleSC/$K_D$ cisSC-PC/PS/PIP$_2$-NDs were 5.91 and 11.09 for WT $C_2AB^*$, were dramatically decreased by the R322E/K325E and R398Q/R399Q mutations, and slightly increased by the E295A/Y338W mutation (*Supplementary file 2*). These findings are fully consistent with the conclusion that interactions of both the polybasic region and R398,R399 with the lipids contribute to Ca$^{2+}$-independent binding of $C_2AB^*$ to cisSC-NDs, and hence that such binding involves at least two types of interactions where either the primary interface binds to the SNAREs and the polybasic region to the lipids, or the polybasic region binds to the SNAREs and R398,R339 to the lipids (*Figure 5—figure supplement 2A,B*). As observed in the previous experiments, the E295A/Y338W mutation increased the former of these two binding modes. As expected, the PC/PS/PIP$_2$-NDs enhanced binding to the SNARE complex more than the PC-NDs, but it is noteworthy that even interactions of $C_2AB^*$ basic residues with neutral phospholipids such as PC can enhance binding to the SNARE complex.

As expected, Ca$^{2+}$ increased the affinity of $C_2AB^*$ for solubleSC, cisSC-PC-NDs and cisSC-PC/PS/PIP$_2$-NDs, and binding was tightest for the latter (*Figure 8A,B*; *Supplementary file 1*). The effects of the R398Q/R399Q, R322E/K325E and E295A/Y338W mutations on Ca$^{2+}$-dependent binding of $C_2AB^*$ to solubleSC and cis-SC-PC-NDs (*Figure 8D,F*) were similar to those observed in the absence of Ca$^{2+}$, and highly efficient FRET was also observed at saturating concentrations, as expected for binding of $C_2AB^*$ to the SNARE complex through the primary or polybasic interfaces with the acceptor probe on residue 76 of SNAP-25 (*Figure 5—figure supplement 1B*). However, the highest FRET efficiency observed upon binding of WT $C_2AB^*$ to cisSC-PC/PS/PIP$_2$-NDs was 0.4 (*Figure 5B*). The highest FRET efficiency was increased to some extent by the E295A/Y338W mutation and even more by the R322E/K325E mutation, but was decreased by the R398Q/R399Q mutation, which consistently led to a bimodal binding curve with a maximum at about 20 nM cisSC-PC/PS/PIP$_2$-NDs in four independent experiments (*Figure 8D,F,H*). These results are consistent with the notion that Ca$^{2+}$-dependent binding of $C_2AB^*$ to PIP$_2$-containing membranes is incompatible with SNARE complex binding through the primary and polybasic interfaces. In the absence of interactions with the SNARE complex, $C_2AB^*$ may bind with similar probabilities to both sides of the nanodiscs, perhaps with some preference for the side lacking SNAREs where there are no steric clashes with the SNARE complex. Non-specific interactions of R398,R399 of WT $C_2AB^*$ with acidic residues from the SNAREs may facilitate binding to the SNARE-containing side (*Figure 5—figure supplement 2E*), leading to some FRET, but mutation of these arginines may lead to preferential binding of the R398Q/R399Q mutant to the other side, particularly at high nanodisc concentrations when membrane availability is not limiting. The highest FRET efficiency was likely increased by the E295A/Y338W mutation because it enhances SNARE binding. High FRET efficiencies were reached for binding of the R322E/K325E mutant to cisSC-PC/PS/PIP$_2$-NDs or for binding of WT $C_2AB^*$ and the three mutants to solubleSC or cisSC-PC-NDs (*Figure 8D,F,H*) because the specific Ca$^{2+}$-dependent interaction of $C_2AB^*$ with PIP$_2$ that competes with SNARE binding is precluded.

# ATP strongly impairs $Ca^{2+}$-dependent binding of $C_2AB^*$ to nanodisc-anchored SNARE complex but $Ca^{2+}$-independent binding persists

Physiological conditions including ATP and $PIP_2$-containing membranes were reported to disrupt Syt1-SNARE complex binding, leading to the conclusion that such binding is not biologically relevant (*Park et al., 2015*). However, a fluorescent probe attached to residue 342 of the $C_2B$ domain to monitor binding to the SNARE complex in this study may have disrupted binding through the primary interface because residue 342 is very close to this interface (*Figure 5—figure supplement 1A*), and another study concluded that Syt1-SNARE complex binding persists in the presence of ATP (*Wang et al., 2016*). To clarify this controversy and examine how ATP affects Syt1-SNARE complex interactions, we first analyzed the effects of adding 2.5 mM $Mg^{2+}$ with or without 2 mM ATP on the FRET observed between $C_2AB^*$ and NDs or cisSC-NDs containing 5% Rho-PE, 15% PS and 1% $PIP_2$. $Mg^{2+}$ and Mg-ATP decreased the FRET with both NDs and cisSC-NDs to some extent in the absence of $Ca^{2+}$, but the FRET with cisSC-NDs was still much stronger that the FRET with NDs (*Figure 9—figure supplement 1*). However, the difference in FRET with cisSC-NDs and NDs was much smaller in the presence of $Ca^{2+}$ when $Mg^{2+}$ and ATP were added (*Figure 9—figure supplement 2*). These results suggested that $Ca^{2+}$-independent binding of $C_2AB^*$ to the SNARE complex persists in the presence of Mg-ATP, but $Ca^{2+}$-dependent binding is strongly impaired. These conclusions were supported by titrations of $C_2AB^*$ with NDs and cisSC-NDs, which revealed a considerable difference in affinity in the absence of $Ca^{2+}$ but a much smaller difference in $Ca^{2+}$-dependent binding (*Figure 9A, B*).

To investigate the nature of the $Ca^{2+}$-independent interaction of $C_2AB^*$ with cisSC-NDs in the presence of Mg-ATP, we analyzed the effects of mutations on $Ca^{2+}$-independent binding to NDs and cisSC-NDs. Binding to cisSC-NDs was enhanced by the E295A/Y338W mutation and markedly impaired by the R398Q/R399Q, R322E/K325E, K324E/K326E and R322E/K325E, R398Q/R399Q mutations, whereas all the mutations had small effects on ND binding (*Figure 9C–D*). The E295A/Y338W mutant bound much tighter to cisSC-NDs than to NDs but all other mutants exhibited similar affinity for NDs and cisSC-NDs (*Figure 9—figure supplement 3*). These results suggest that, in the presence of Mg-ATP, $Ca^{2+}$-independent binding of WT $C_2AB^*$ to the SNARE complex on nanodiscs also involves at least two binding modes mediated by either the primary or the polybasic interface, and the former is stabilized by the E295A/Y338W mutation. We also analyzed whether complexin-1 altered the affinity of WT $C_2AB^*$ for cisSC-NDs, but did not observe any significant effects (*Figure 9E*). We also verified that the R322E/K325E mutation disrupts $Ca^{2+}$-dependent binding to $PIP_2$-containing NDs much more strongly than the K324E/K326E mutation in the presence of Mg-ATP (*Figure 9F*), confirming the specificity of such impairment under these conditions.

## Discussion

Because of the well-established functions of the SNAREs as the engines of membrane fusion and of Syt1 as the $Ca^{2+}$ sensor that triggers synchronous neurotransmitter release, elucidating the role(s) of Syt1-SNARE interactions is crucial to understand how $Ca^{2+}$ sensing is coupled to membrane fusion during release. The recent determination of three structures of Syt1-SNARE complexes led to intriguing models of $Ca^{2+}$-triggered release (*Brewer et al., 2015*; *Zhou et al., 2015*; *Zhou et al., 2017*), but yielded a confusing picture because of the striking differences among the structures. Paradoxically, $Ca^{2+}$ was generally believed to enhance Syt1-SNARE binding, but the $C_2$ domain $Ca^{2+}$-binding loops were not involved in SNARE binding in any of the structures. Together with available data, the study presented here suggests that Syt1 binds to the SNARE complex before $Ca^{2+}$ influx, most likely through the primary interface, and that $Ca^{2+}$ actually releases this interaction, inducing tight membrane binding that involves specific interactions of the $C_2B$ polybasic region with $PIP_2$. We propose that the Syt1-SNARE complex keeps the release machinery in a state that hinders membrane fusion but at the same time is ready for fast release when $Ca^{2+}$ induces binding of the Syt1 $Ca^{2+}$-binding loops to the membrane, releasing the Syt1-SNARE interaction and enabling cooperation between Syt1 and the SNAREs in membrane fusion (*Figure 10*).

To rationalize the immense amount of data available on Syt1-SNARE interactions, it is crucial to decipher which data reflect physiologically relevant interactions and which arise from the promiscuity of these proteins, the lack of key components in a reduced system and/or the choice of experimental

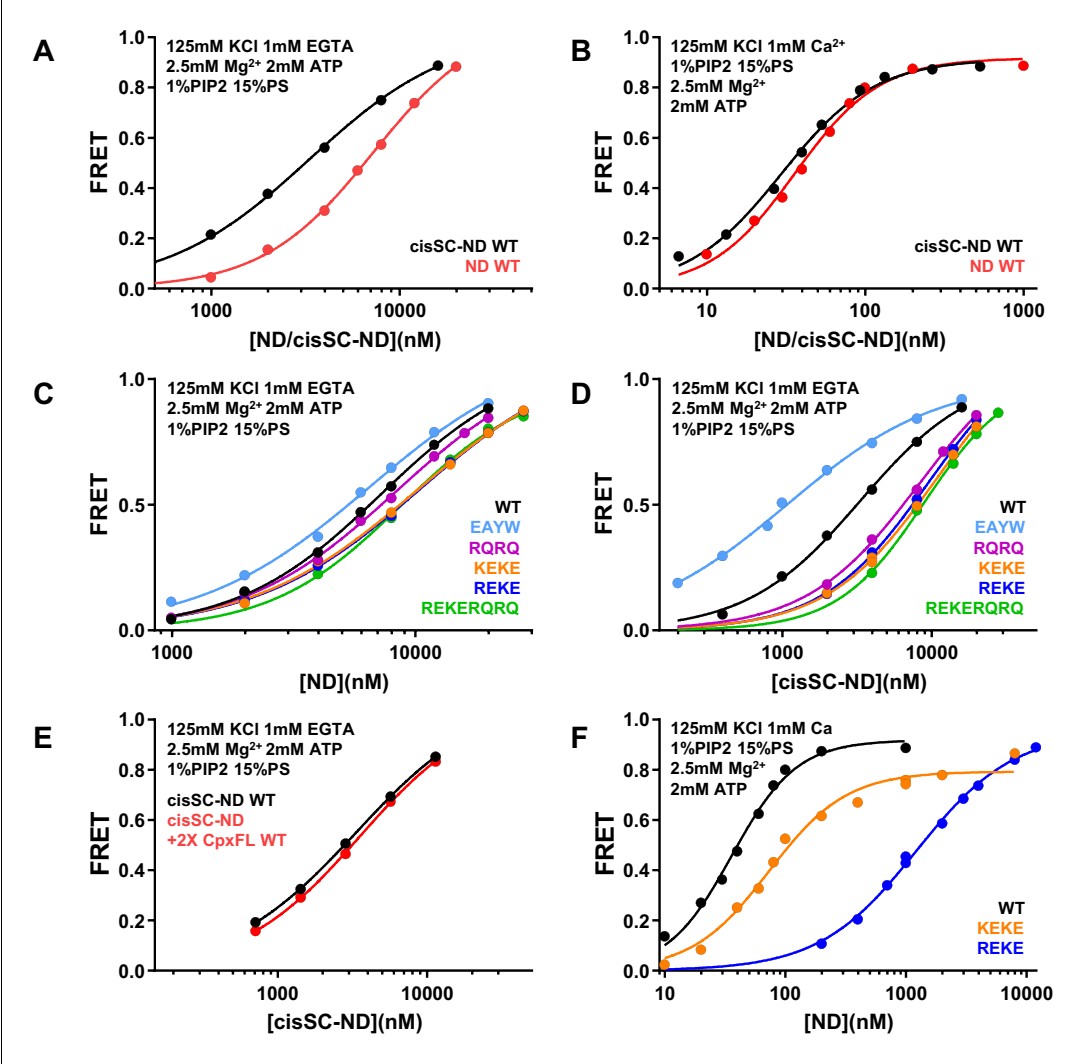

**Figure 9.** ATP almost abolishes $Ca^{2+}$-dependent binding but not $Ca^{2+}$-independent binding of $C_2AB*$ to SNARE complexes in cisSC-NDs. (**A–B**) FRET efficiencies observed in titrations of WT $C_2AB*$ with NDs or cisSC-NDs containing 15% PS, 5% Rho-PE and 1% $PIP_2$ in the presence of 2.5 mM $Mg^{2+}$ and 2 mM ATP, and 1 mM EGTA plus 50 mM KCl (**A**), or 1 mM $Ca^{2+}$ plus 125 mM KCl (**B**). (**C–D**) FRET efficiencies observed in titrations of WT, E295A/Y338W (EAYW), R398Q/R399Q (RQRQ), R322E/K325E (REKE), K324E/K326E (KEKE) or R322E/K325E/R398Q/R399Q (REKERQRQ) $C_2AB*$ with NDs (**C**) or cisSC-NDs (**D**) containing 15% PS, 5% Rho-PE and 1% $PIP_2$ in the presence of 2.5 mM $Mg^{2+}$, 2 mM ATP, 1 mM EGTA and 50 mM KCl. (**E**) FRET efficiencies observed in titrations of $C_2AB*$ with cisSC-NDs containing 15% PS, 5% Rho-PE and 1% $PIP_2$ in the presence of 2.5 mM $Mg^{2+}$, 2 mM ATP, 1 mM EGTA and 50 mM KCl, with or without complexin-1. (**F**) FRET efficiencies observed in titrations of WT, R322E/K325E (REKE) or K324E/K326E (KEKE) $C_2AB*$ with NDs containing 15% PS, 5% Rho-PE and 1% $PIP_2$ in the presence of 2.5 mM $Mg^{2+}$, 2 mM ATP, 1 mM $Ca^{2+}$ and 125 mM KCl.

The online version of this article includes the following source data and figure supplement(s) for figure 9:

**Source data 1.** Source data for *Figure 9*.

**Figure supplement 1.** ATP does not abolish $Ca^{2+}$-independent binding of $C_2AB*$ to SNARE complexes in cisSC-NDs.

**Figure supplement 1—source data 1.** Source data for *Figure 9—figure supplement 1*.

**Figure supplement 2.** ATP does almost abolishes $Ca^{2+}$-dependent binding of $C_2AB*$ to SNARE complexes in cisSC-NDs.

**Figure supplement 2—source data 1.** Source data for *Figure 9—figure supplement 2*.

**Figure supplement 3.** Effects of mutations in the Syt1 $C_2B$ domain on $Ca^{2+}$-independent binding of $C_2AB*$ to NDs and cisSC-NDs in the presence of ATP.

**Figure supplement 4.** Reproducibility of titrations of $C_2AB*$ with NDs and cisSC-NDs.

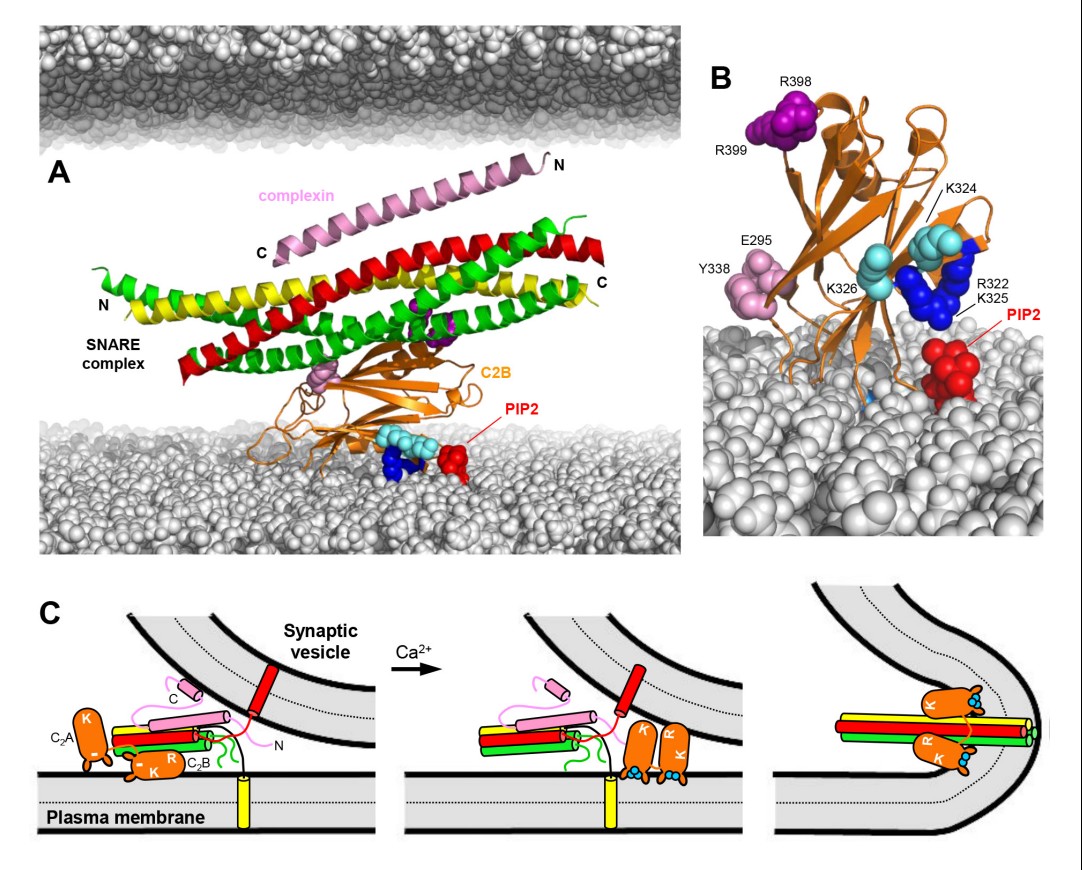

**Figure 10.** Working model for the function of Syt1 in neurotransmitter release. (**A**) Model of how the Syt1 $C_2B$ domain (orange) can bind to the SNARE complex (syntaxin-1 yellow, SNAP-25 green, synaptobrevin red) through the primary interface and at the same time to the plasma membrane through the polybasic region. Complexin-1 (26–83) bound to the other side of the SNARE complex is also shown (pink). The N- and C-termini of the SNARE for-helix bundle and complexin-1 (26–83) are indicated. The plasma membrane is shown below and the vesicle membrane above (both in gray). The model shows how, in this orientation, K324 and K326 (cyan spheres) on one side of a β-strand are readily accessible for binding to $PIP_2$ (red) on the plasma membrane, similar to R322 and K325 (dark blue spheres) on the other side of the same β-strand. R398 and R399 are shown as purple spheres, and E295 and Y338 as pink spheres. The model also shows how this binding mode hinders C-terminal zippering of the SNARE complex because the C-terminus of the syntaxin-1 SNARE motif would be far from the plasma membrane. Complexin-1 also hinders zippering because its accessory helix bumps with the vesicle membrane. The model was constructed by superimposing the structures of the complexes formed by the SNARE complex with the $C_2B$ domain and complexin-1 (26–83) (PDB accession codes 5kj7 and 1KIL). (**B**) Model showing how $Ca^{2+}$-induced insertion of the Syt1 $C_2B$ domain $Ca^{2+}$-binding loops into the plasma membrane is expected to yield an approximately perpendicular orientation that allows binding of R322 and K325 to $PIP_2$, whereas K324 and K326 cannot reach $PIP_2$ because they point in the opposite direction. (**C**) Proposed model of neurotransmitter release whereby the Syt1 $C_2B$ domain is bound to the SNARE complex and the plasma membrane before $Ca^{2+}$ influx, while complexin-1 is bound on the other side (as shown in more detail in panel **A**). The N- and C-terminal sequences of complexin-1 are shown as pink curves that represent unfolded structure, except for a helix at the C-terminus that is believed to bind to the vesicle membrane (*Snead et al., 2014*). The location of the Syt1 $C_2A$ domain is arbitrary. The polybasic region of $C_2B$ is indicated by K, while R398,R399 at the bottom of $C_2B$ is indicated by an R. A basic sequence at the bottom of $C_2A$ is indicated by K. The model proposes that the inhibition of membrane fusion in this state is rapidly relieved by $Ca^{2+}$- and $PIP_2$-dependent binding of both Syt1 $C_2$ domains to the plasma membrane, releasing the interaction with the SNAREs and bridging the two membranes (middle panel) to facilitate C-terminal SNARE zippering ($Ca^{2+}$ ions shown as blue circles). Induction of positive curvature by insertion of the $C_2$ domain $Ca^{2+}$-binding loops into the membranes may cooperate with SNARE zippering in catalyzing membrane fusion (right panel). Note that the curvature of the membrane fusion diagram in the plane perpendicular to the paper is positive.

conditions. This task is hindered by the fact that relevant interactions may be necessarily weak because of the very nature of this dynamic system, which is expected to undergo quick, drastic rear-rangements during the events that lead to fast membrane fusion upon $Ca^{2+}$ influx. Weak interactions can be dramatically enhanced by co-localization within the confines of a primed synaptic vesicle, but they must be distinguished from other, perhaps stronger but irrelevant interactions that are detect-able in vitro (*Magdziarek et al., 2020*). Another complicating aspect is that basic sequences such as

the polybasic region and R398,R399 can bind to SNAREs and to membranes, both of which are acidic, and binding to a wrong molecule can occur in the absence of the bona fide target. Perhaps the most confusing factor was the observation by multiple labs that $Ca^{2+}$ strongly enhanced binding of Syt1 to SNAREs or SNARE complexes in solution (see introduction), leading to the widespread belief that elucidating how $Ca^{2+}$-bound Syt1 binds to the SNARE complex was a 'Holy Grail' to understand neurotransmitter release. However, it now seems clear that $Ca^{2+}$ releases Syt1 from SNARE complexes anchored on $PIP_2$-containing membranes such as the plasma membrane and that the $Ca^{2+}$-induced increase in Syt1-SNARE binding in solution arose merely because $Ca^{2+}$ increases the positive electrostatic potential of the Syt1 $C_2$ domains (*Fernandez et al., 2001*; *Shao et al., 1997*). This enhanced Syt1-SNARE affinity is offset by the specific, $Ca^{2+}$-dependent interaction of Syt1 with $PIP_2$-containing membranes, which is incompatible with SNARE complex binding and is much stronger. This key conclusion arises from several lines of evidence.

First, our titrations with NDs and cisSC-NDs show that physiological conditions including Mg-ATP almost abolish $Ca^{2+}$-dependent binding of $C_2AB^*$ to the SNARE complex anchored on $PIP_2$-containing membranes (*Figure 9B*), consistent with previous results (*Park et al., 2015*). A previous study reported that Syt1-SNARE complex interactions persist in the presence of ATP (*Wang et al., 2016*), but this conclusion was tested only in the absence of $Ca^{2+}$ and is thus consistent with our results (*Figure 9A*).

Second, and particularly important is the observation of the high specificity of $Ca^{2+}$-dependent binding of $C_2AB^*$ to $PIP_2$-containing NDs, which is disrupted much more strongly by the R322E/K325E mutation than by the K324E/K326E mutation (*Figures 6G* and *9F*), in correlation with the effects of these mutations on neurotransmitter release (*Brewer et al., 2015*). Note that, remarkably, the striking difference in the effects of the two mutations on $Ca^{2+}$-dependent binding of $C_2AB^*$ to $PIP_2$-containing NDs was not observed in $Ca^{2+}$-dependent binding to NDs lacking $PIP_2$ (*Figure 6E*) or $Ca^{2+}$-independent binding to NDs with or without $PIP_2$, and with or without SNARE complex (*Figure 6A–D*). As explained above, modeling readily explains these observations, as R322 and K325 (but not K324 and K326) are expected to be well positioned to interact with $PIP_2$ upon $Ca^{2+}$-induced binding of the $C_2B$ domain to the membrane (*Figure 10B*). In the absence of $Ca^{2+}$, insertion of the $Ca^{2+}$-binding loops into the membrane is hindered by the negative charge of the loops (*Fernandez et al., 2001*) and, consequently, more parallel orientations of the $C_2B$ domain with respect to the membrane that allow simultaneous binding of the primary interface to the SNARE complex (*Figure 10A*) are favored. These parallel orientations bring K324 and K326 near the membrane and hence $PIP_2$ can readily interact with these residues as well as with R322 and K325.

Third, binding of the $C_2B$ domain to the membrane in the perpendicular orientation induced by $Ca^{2+}$ and $PIP_2$ is incompatible with the binding modes observed in the three structures of Syt1-SNARE complexes that have been determined. The elongated SNARE complex could not remain membrane-anchored or would have strong steric clashes if it remained bound to the $C_2B$ domain through the primary or tripartite interfaces, and binding of the SNARE complex to the polybasic region is hindered because this region interacts with $PIP_2$. These conclusions are supported by the observation that highly efficient FRET between $C_2AB^*$ and the SNARE complex labeled at residue 76 of SNAP-25 was observed under a variety of conditions except for those that allowed $Ca^{2+}$-dependent binding of $C_2AB^*$ to $PIP_2$-containing SNARE complex nanodiscs (*Figure 8*). Note also that the physiological relevance of our NMR structure was supported by the differential disruption of $C_2B$-SNARE complex binding and neurotransmitter release caused by the R322E/K325E and K324E/K326E mutations (*Brewer et al., 2015*). However, the physiological data can now be explained by the differential effects of these mutations on $Ca^{2+}$-dependent binding of $C_2AB^*$ to $PIP_2$-containing membranes (*Figures 6G* and *9F*), which occurs with high affinity (in the low nM range) and is much tighter than SNARE complex binding (*Figure 8D*).

Syt1 does bind to membrane-anchored SNARE complexes in the absence of $Ca^{2+}$, even in the presence of ATP (*Figure 9A*; *Wang et al., 2016*). Our NMR studies show that there are two main $Ca^{2+}$-independent binding modes between the $C_2B$ domain and CpxSC in solution (*Figure 1*) that are mediated by the primary and polybasic interfaces. The strong disruption of CpxSC binding to the primary interface caused by the R398Q/R399Q mutation (*Figure 2B,C*) shows that R398,R399 contribute substantially to the energy of binding at this interface. E295 and Y338 likely contribute also to the binding energy, but additional mutations will be necessary

to assess this contribution, as the E295A/Y338W mutation actually enhances binding (*Figure 4*). These observations contrast with the finding that co-IP of Syt1 with the SNAREs was not substantially affected by the R398Q/R399Q mutation and was moderately disrupted by the E295A/Y338W mutation (*Zhou et al., 2015*). This discrepancy might arise because co-IP likely detects SNARE binding to the polybasic region upon membrane solubilization, and depends not only on affinities but also on off rates. The effects of the R398Q/R399Q and E295A/Y338W mutations observed by NMR correlate with those observed on $Ca^{2+}$-independent binding of $C_2AB^*$ to cisSC-NDs (*Figures 6B,D*, *7A*, *8* and *9D*), showing that binding to the SNARE complex on the nanodiscs is mediated in part by the primary interface. However, the R322E/K325E and K324E/K326E mutations also decreased the differences in the affinities of $C_2AB^*$ for NDs and cisSC-NDs (*Figure 6—figure supplements 1–3*; *Figure 9—figure supplement 3*), indicating that there are additional binding modes where the $C_2B$ polybasic region interacts with the SNAREs. These other binding modes are unlikely to be biologically relevant, as the K324E/K326E mutation does not impair neurotransmitter release substantially (*Brewer et al., 2015*).

The functional importance of binding of Syt1 to the SNARE complex through the primary interface is overwhelmingly supported by physiological data (*Guan et al., 2017*; *Zhou et al., 2015*). Hence, it is most likely that binding of $C_2AB^*$ to the SNARE complex in cisSC-NDs through the polybasic region arises from limitations of our in vitro experiments and that Syt1 is bound to the SNARE complex through the primary interface before $Ca^{2+}$ influx. Although this interaction is rather weak, it is likely to be dramatically stabilized by co-localization at the site of fusion and may be favored over other binding modes by other factors. For instance, our experiments used 1% $PIP_2$, which corresponds to the average $PIP_2$ content of the plasma membrane, but clusters containing 6% $PIP_2$ have been detected (*James et al., 2008*; *van den Bogaart et al., 2011*). High $PIP_2$ concentration should favor simultaneous binding of the $C_2B$ domain polybasic region to this lipid and of the primary interface to the SNARE complex (*Figure 10A,C*). Such a state has been observed by cryo-EM on lipid nanotubes, albeit at limited resolution (*Grushin et al., 2019*). The structure revealed a slanted orientation of the SNARE complex that should hinder full zippering at the C-terminus and was disrupted by $Ca^{2+}$. It is unclear whether the orientation was dictated by helical packing on the nanotubes and whether $Ca^{2+}$ disrupted such packing, but these results are consistent with the notion that $Ca^{2+}$ releases the interaction between Syt1 and membrane-anchored SNARE complexes. Even if the SNARE complex is less slanted in the primed state, a more parallel orientation of the SNARE complex with respect to the membrane (e.g. *Figure 10A*) would still hinder C-terminal zippering, which could explain the increase in spontaneous release observed in Syt1 KO mice (*Xu et al., 2009*). The proposed state is also consistent with the observation that complexin-1 does not alter the affinity of $C_2AB^*$ for cisSC-NDs (*Figure 9E*), as complexin-1 and Syt1 bind to opposite sides of the SNARE complex (*Figure 10A*). In this state, the helix formed by complexin-1 would point toward the vesicle membrane (*Figure 10A,C*), also hindering SNARE complex zippering (*Trimbuch et al., 2014*). This model provides a basis for the findings that complexin-1 is required for the dominant negative effect of Syt1 bearing mutations in the $C_2B$ domain $Ca^{2+}$-binding sites (*Zhou et al., 2017*) and that Syt1 is necessary for the inhibition of spontaneous release by complexin in *Drosophila* (*Jorquera et al., 2012*).

Our finding that the R398Q/R399Q mutation strongly disrupts binding of the $C_2B$ domain to the SNARE complex through the primary interface (*Figure 2B,C*) suggests that such disruption underlies the dramatic impairment of $Ca^{2+}$-evoked neurotransmitter release caused by this mutation, as well as the abrogation of clamping of spontaneous release by Syt1 (*Xue et al., 2008*; *Zhou et al., 2015*). The E295A/Y338W mutation in the primary interface also disrupted neurotransmitter release but still allowed clamping of spontaneous release, which resembles the phenotype observed for the R322E/K325E mutation in the polybasic region (*Zhou et al., 2015*). These findings are readily explained by our model and our biochemical data, as the E295A/Y338W enhances (rather than weakens) $Ca^{2+}$-independent binding of Syt1 to the SNARE complex (*Figures 4*, *7A*, *8* and *9D*) and hence hinders release of the inhibitory Syt1-SNARE interaction. Similarly, the R322E/K325E mutation impairs $Ca^{2+}$- and $PIP_2$-dependent membrane binding, which is critical to release this inhibitory interaction.

Altogether, the available data support the notion that, before $Ca^{2+}$ influx, the trans-SNARE complex, complexin-1 and Syt1 form a macromolecular assembly between the vesicle and plasma membranes that inhibits release but is ready to trigger fast membrane fusion upon $Ca^{2+}$ influx because it

brings Syt1 close to the SNAREs (*Figure 10C* left panel). In this model, $Ca^{2+}$ triggers tight, specific binding of the $C_2B$ domain to $PIP_2$ and other lipids in the plasma membrane while binding of $C_2B$ to the SNARE complex is released. The mechanisms underlying the last events leading to membrane fusion are still unclear, but we speculate that the Syt1 $C_2$ domains bridge the vesicle and plasma membranes (*Araç et al., 2006*; *Figure 10C*, middle panel). This action could cooperate with the SNAREs in bringing the membranes together, and insertion of the $C_2$ domain $Ca^{2+}$-binding loops into the membrane can induce membrane curvature (*Martens et al., 2007*) to facilitate membrane fusion (*Figure 10C*, right panel). Note that R398,R399 are critical for the membrane-membrane bridging activity of Syt1 and therefore the strong disruption of neurotransmitter release caused by mutating these arginines might arise from impairment of this activity (*Araç et al., 2006*; *Xue et al., 2008*) in addition to abrogating SNARE binding.

Clearly, the proposed model needs further testing and multiple aspects need to be unraveled to understand the mechanism of neurotransmitter release. Thus, syntaxin-1 is known to form clusters on the plasma membrane (*Khuong et al., 2013*), and formation of Syt1 oligomers has been proposed to underlie the primed state and to exert an inhibitory activity that is released by $Ca^{2+}$ (*Wang et al., 2017*). Such clusters and oligomers could not be present in our nanodisc experiments. However, the Syt1 oligomers are compatible with the state proposed in *Figure 10A* (*Tagliatti et al., 2020*) and hence can be readily incorporated into the model of *Figure 10C*. Note also that, while our NMR studies could not detect binding of the Syt1 $C_2B$ domain to CpxSC through the tripartite interface (*Figures 2E* and *3*), it is premature to completely rule out the relevance of the tripartite structure given the potential functional importance of very weak interactions (*Magdziarek et al., 2020*). Nevertheless, it is also premature to accept this structure as physiologically relevant, and further evidence for such relevance needs to be obtained, ideally using mutations that exclusively replace residues in the interface rather than interior residues that are important for protein stability. The model of *Figure 10* provides a strong foundation to address these issues and incorporate additional proteins that may play key roles in the last steps of neurotransmitter release.

# Materials and methods

**Key resources table**

| Reagent type (species) or resource | Designation | Source or reference | Identifiers | Additional information |
|---|---|---|---|---|
| Recombinant DNA reagent | pGEX-KG-GST-Syt1_C2B | *Brewer et al., 2015* | | Protein expression plasmid for *E. coli* (rat synaptotagmin-1 $C_2AB$ domain, residues 271–421) |
| Recombinant DNA reagent | pGEX-KG-GST-Syt1_C2AB | *Araç et al., 2006* | | Protein expression plasmid for *E. coli* (rat synaptotagmin-1 $C_2AB$ domain, residues 140–421) |
| Recombinant DNA reagent | pMSP1E3D1 | Addgene | 20066 | Protein expression plasmid for *E. coli* (MSP1E3D1) |
| Chemical compound, drug | DEUTERIUM OXIDE (D, 99.8%) | Cambridge Isotope Laboratories, Inc | DLM-4–99.8 | |
| Chemical compound, drug | AMMONIUM CHLORIDE ($^{15}N$, 99%) | Cambridge Isotope Laboratories, Inc | NLM-467 | |
| Chemical compound, drug | D-GLUCOSE (1,2,3,4,5,6,6-$D_7$, 97–98%) | Cambridge Isotope Laboratories, Inc | DLM-2062 | |
| Chemical compound, drug | L-Methionine-(methyl-$^{13}C$) | Sigma-aldrich | 299146 | |

*Continued on next page*

*Continued*

| Reagent type (species) or resource | Designation | Source or reference | Identifiers | Additional information |
|---|---|---|---|---|
| Chemical compound, drug | 2-Ketobutyric acid -4–$^{13}$C,3,3-d$_2$ sodium salt hydrate | Sigma-aldrich | 589276 | |
| Chemical compound, drug | PC | Avanti polar lipid | 850457 | 1-palmitoyl-2-oleoyl-glycero-3-phosphocholine |
| Chemical compound, drug | PS | Avanti polar lipid | 840035 | 1,2-dioleoyl-sn-glycero-3-phospho-L-serine (sodium salt) |
| Chemical compound, drug | PIP2 | Avanti polar lipid | 840046 | L-$\alpha$-phosphatidylinositol-4,5-bisphosphate (Brain, Porcine) (ammonium salt) |
| Chemical compound, drug | RhoPE | Avanti polar lipid | 810158 | 1,2-dipalmitoyl-sn-glycero-3-phosphoethanolamine-N-(lissamine rhodamine B sulfonyl) (ammonium salt) |
| Chemical compound, drug | Tetramethylrhodamine-5-Maleimide | Invitrogen | T-6027 | |
| Chemical compound, drug | Alexa Fluor488 C5 Maleimide | Thermo Fisher Scientific | A10254 | |
| Chemical compound, drug | $\beta$-OG (octyl-beta-glucoside) | Gold biotechnology | O-110–50 | |
| Chemical compound, drug | Detergent Removal Resin | Thermo Fisher Scientific | 87780 | |

## Protein expression and purification

Constructs to express the following proteins or protein fragments were described previously: rat synaptobrevin-2 SNARE motif (residues 29–93), rat synaptobrevin-2 (residues 49–93), human SNAP-25A fragments encoding its SNARE motifs (residues 11–82 and 141–203), full-length rat synaptobrevin, full-length rat syntaxin-1A, rat syntaxin-1A (residues 191–253), rat syntaxin-1A (residues 2–253), human SNAP-25A (residues 11–82 and 141–203), rat synaptotagmin-1 C$_2$B domain with a C277A mutation (residues 271–421; referred to as WT C$_2$B domain), the same rat synaptotagmin-1 fragment with a R398Q/R399Q mutation, rat synaptotagmin-1 C2AB fragment with a C277A mutation (residues 140–421), full-length rat complexin-1, rat complexin-1 (residues 26–83) and MSP1E3D1 (pMSP1E3D1 was a kind gift from Stephen Sligar; Addgene plasmid # 20066; http://n2t.net/addgene:20066; RRID:Addgene_20066) (*Araç et al., 2006*; *Brewer et al., 2015*; *Chen et al., 2006*; *Chen et al., 2008*; *Chen et al., 2002*; *Denisov et al., 2007*; *Ma et al., 2013*; *Xu et al., 2013*; *Zhou et al., 2013*). All these proteins were expressed in *E. coli* BL21 (DE3) cells and purified as previously described in these references, with the following exceptions.

The synaptobrevin-2(29–93), syntaxin-1A (191–253), SNAP-25A(11–82), SNAP-25A(141–203) and complexin-1(26–83) fragments were expressed *E. coli* BL21 (DE3) cells with the pET-duet vector. Both SNAP-25 fragments included tryptophan at the N-terminus to facilitate detection by UV spectroscopy. Cells were harvested and re-suspended in PBS pH 7.4 with 10 mM imidazole and supplemented with Sigma protease inhibitors (P2714-1BTL). Cleared lysates were applied to Ni-NTA resin (Thermo Fisher), washed with PBS pH 7.4 with 10 mM imidazole, PBS pH 7.4 with 10 mM imidazole and 10% Triton, PBS pH 7.4 with 10 mM imidazole and 1 M NaCl, and eluted in PBS pH 7.4, 500 mM imidazole. The affinity tag was cleaved with TEV protease overnight at 4°C in PBS pH 7.4. After affinity tag cleavage, all proteins were further purified using size exclusion chromatography on a Superdex 75 column (GE 16/60) equilibrated with 20 mM Tris pH 7.4 125 mM NaCl. MSP1E3D1 was expressed in *E. coli* BL21 (DE3) cells grown in Terrific broth media to OD600 = 2.0, then induced with 1 mM Isopropyl$\beta$-D-1-thiogalactopyranoside (IPTG) for 4 hr at 37°C. Cells were re-suspended in 40 mM Tris pH 8.0 300 mM NaCl 1% TritonX-100 5 mM Imidazole containing Sigma protease

inhibitors (P2714-1BTL) and lysed using an Avestin EmulsiFlex-C5 homogenizer. The soluble fraction of the cell lysate was collected after centrifugation at 48,000 x g for 45 min and incubated with Ni-NTA resin (Thermo Fisher) for 2 hr at 4°C. The resin was washed with the following buffers sequentially: 40 mM Tris pH 8.0 300 mM NaCl 1% Triton, 40 mM Tris pH 8.0 300 mM NaCl 50 mM Sodium Cholate 20 mM Imidazole and 40 mM Tris pH 8.0 300 mM NaCl 50 mM Imidazole followed by elution using 40 mM Tris pH 8.0 300 mM NaCl 500 mM imidazole. This was followed by size exclusion chromatography on a Superdex 200 column (GE 16/60) in 20 mM Tris pH 7.4 100 mM NaCl 0.5 mM EDTA.

Expression vectors for mutant proteins were generated using a combination of the QuickChange site-directed mutagenesis kit (Strategene) and standard PCR-based techniques with custom designed primers. These mutants included: Syt1 $C_2B$ (residues 271–421) E295A/Y338W, L387Q/L394Q, R398Q, R399Q, R322E/K325E, and R322E/K325E/R399Q/R398Q; Syt1 $C_2AB$ (residues 140–421) C277A/E346C (referred to as WT $C_2AB^*$), C277A/E346C/R398Q/R399Q, C277A/E346C/R322E/K325E, C277A,E346C,R322E/K325E/R398Q/R399Q, C277A/E346C/E295A/Y338W and C277A/E346C/K324E/K326E; and SNAP-25 (residues 11–82) R16C, Q34C and K76C. All mutant proteins were purified as the WT proteins, including 0.5 mM TCEP in the final purification step for cysteine containing proteins. Isotopically labeled and perdeuterated Syt1 $C_2B$ domain and Cpx-1 (26–83) fragments were expressed using M9 expression media in 99.9% $D_2O$ with D-glucose (1,2,3,4,5,6,6-$D_7$, 97–98%) as the sole carbon source (3 g/L) and $^{15}NH_4Cl$ as the sole nitrogen source (1 g/L). Specific $^{13}CH_3$-labeling at the Met and Ile $\delta1$ methyl groups of Syt1 $C_2B$ domain fragments was achieved by adding [3,3–$^2H$] $^{13}C$-methyl $\alpha$-ketobutyric acid (80 mg/L) and 13C-methyl methionine (250 mg/L) (Cambridge Isotope Laboratories) to the cell cultures 30 min prior to IPTG induction.

## Assembly of soluble SNARE complex for NMR experiments

The SNARE motifs were mixed in the equimolar ratio in following order: synaptobrevin-2 (29–93), SNAP-25A(141–203), SNAP-25A(11–82) and syntaxin-1A (191–253), in the presence of 1 M NaCl. The mixture contained the following protease inhibitors (protease inhibitor cocktail A): Antipain Dihydrochloride 0.016 mg/ml (Thermo Fischer Scientific: 50488492); Leupeptin 0.33 mg/ml (Gold Bio: L01025); Aprotinin 0.08 mg/ml (Gold Bio: A655100). The assembly reaction was incubated in room temperature overnight while rotating. The SNARE motifs that did not incorporate into complex were removed by concentration-dilution at room temperature using 30 kDa molecular weight cutoff (MWCO) Amicon centrifugation filters.

## Syt1 $C_2B$ domain/SNARE complex solubility tests

Samples containing 80 µM WT Syt1 $C_2B$ domain were mixed with 30 µM SNARE complex +/- Complexin-1 (26–83) in 20 mM HEPES, 125 mM KCl, 1 mM $CaCl_2$, pH 7.4. The total reaction volume was 30 µl. After mixing, the UV spectra of both samples were acquired using Nanodrop (Thermofisher). Next, the samples were centrifuged at 13,000 rpm for 5 min spin in a benchtop centrifuge (Eppendorf). The UV spectra of both samples were acquired again after centrifugation.

## NMR spectroscopy

All NMR spectra were acquired at 25°C on Agilent DD2 spectrometers operating at 600 or 800 MHz and equipped with cold probes. $^1H$-$^{15}N$ TROSY HSQC and $^1H$-$^{13}$ HMQC spectra were acquired on samples with 10% $D_2O$ as the solvent. For titrations of WT and mutant Syt1 $C_2B$ domains with SNARE-Complexin-1 (26–83) complex, assembled soluble SNARE complex was concentrated at room temperature to a concentration above 250 µM using 30 kDa (Amicon) centrifugation filters and buffer exchanged into 20 mM HEPES containing 1 mM TCEP and 100 or 125 mM KCl 10% D2O buffer using Zeba Spin Desalting Columns, 7K MWCO, 10 mL (Thermofisher). Complexin-1 (26–83) was also concentrated above 250 µM using 3 kDa (Amicon) centrifugation filters and buffer exchanged to matching buffer. SNARE-Complexin-1 (26–83) complex was preassembled before mixing with $C_2B$ Syt-1 fragment with 20% excess Complexin-1 (26–83), and 1 mM EDTA or 1 mM $CaCl_2$ were included in the mix depending on the final sample buffer. After cation exchange, the corresponding $^2H,^{15}N$-IM-$^{13}CH_3$-$C_2B$ domain fragment was buffer exchanged using a PD-10 Desalting Column (GE Healthcare) to matching buffer. Final samples containing 32 µM $C_2B$ domain were prepared by adding 1 mM EDTA or 1 mM CaCl2 and protease inhibitor cocktail A. $^1H$-$^{15}N$ TROSY-

HSQC and $^{1}$H-$^{13}$C HMQC spectra were acquired first for the isolated C$_2$B domain and then with the same sample after adding increasing concentrations of complexin-1 (26–83)/SNARE complex. The concentrations for each titration step for each mutant are indicated in the figure legends. For titrations of $^{2}$H,$^{15}$N-CpxSC with WT and R322E/K325E/R398Q/R399Q mutant Syt1 C$_2$B domains, the proteins were prepared by the same procedures described above. $^{2}$H,$^{15}$N-CpxSC was formed with $^{2}$H,$^{15}$N-Cpx1(26-83) and 12.5% excess assembled soluble SNARE complex. $^{1}$H,$^{15}$N TROSY-HSQC spectra were acquired for $^{2}$H,$^{15}$N-CpxSC alone and for 40 μM $^{2}$H,$^{15}$N-CpxSC in the presence of 40 μM WT or R322E/K325E/R398Q/R399Q mutant Syt1 C$_2$B domain in 20 mM HEPES 100 mM KCl 1 mM EDTA 1 mM TCEP. All NMR samples included protease inhibitor cocktail A. Total acquisition times ranged from 3.5 to 87.5 hr, depending on the sensitivity of the spectra. NMR data were processed with NMRPipe (*Delaglio et al., 1995*) and analyzed with NMRView (*Johnson and Blevins, 1994*).

## Labeling proteins with Alexa fluor 488 and tetramethylrhodamine (TMR)

Single cysteine mutants of Synaptotagmin-1 C$_2$AB (140-421) were labeled with Alexa488 using maleimide reactions with Alexa Fluor488 C5 Maleimide reagent from Thermo Fisher Scientific (A10254). Synaptotagmin-1 C$_2$AB single cysteine mutants were first buffer exchanged into 20 mM HEPES pH 7.4 125 mM KCl 0.5 mM TCEP using concentration and dilution. Buffer exchanged proteins at a concentration of 100–150 μM were incubated with 10-fold excess dye for 4 hr at room temperature or overnight at 4˚C. The reaction was quenched by addition of 10 mM DTT. Unreacted dye was separated from labeled protein through cation exchange chromatography on a HiTrap SP column (GE) in 50 mM Sodium Acetate pH 6.2 (Buffer A) using a linear gradient from 0 to 1000 mM NaCl. Prior to elution, the column with the bound protein was washed with 100 mL of buffer A. Single cysteine mutants of SNAP-25A(11-82) were labeled with TMR using maleimide reactions with TMR-5-Maleimide reagent from Thermo Fisher Scientific (T6027). Proteins were first buffer exchanged into 20 mM HEPES pH 7.4 125 mM KCl 0.5 mM TCEP using concentration dilution. Buffer exchanged proteins at a concentration of 100–150 μM were incubated with 10-fold excess dye for 4 hr at room temperature or overnight at 4˚C. The reaction was quenched by addition of 10 mM DTT. Unreacted dye was separated from the labeled protein by using two runs on PD Miditrap G25 column followed by size exclusion chromatography on Superdex 75 column (GE 10/300 or 16/60). The concentration and the labeling efficiency of the tagged proteins were determined by using UV-vis absorbance and a Bradford assay.

## Preparation of nanodiscs, cisSC-nanodiscs and transSC-nanodiscs

Appropriate lipid mixtures (specific to each experiment as indicated in the figures and *Supplementary file 1*) were prepared by mixing chloroform stocks in glass test tubes. These mixtures were dried under a stream of nitrogen and stored overnight in a vacuum desiccator. The lipids were solubilized in a 20 mM HEPES pH 7.4 125 mM KCl 1% β-OG (octyl-beta-glucoside) buffer by vortexing for 5 min. To form nanodiscs, MSP1E3D1 was incubated with solubilized lipids at a ratio of 1:110 in the presence of 1% β-OG (final concentration) at 4˚C for 30 min. The mixture was passed over a 4 cm-high Thermo Scientific Pierce Detergent Removal Resin (87780) column (approximately 3 mL of the slurry; the final volume of the mixture was always between 3–4 mL). The nanodiscs were further purified by size exclusion chromatography using a Superdex 200 column (GE 16/60). Appropriate fractions were pooled and concentrated to a desired concentration using a 30 kDa MWCO Amicon centrigufation filter.

To prepare cisSC-NDs, detergent solubilized cisSNARE-complex was formed by incubating 10 μM full-length rat syntaxin-1A with 15 μM of synaptobrevin-2 (29–93), SNAP-25A(11-82) and SNAP-25A(141-203) in the presence of 1% β−OG, 0.5 M NaCl and protease inhibitor cocktail A overnight at 4˚C. To obtain transSC-NDs, we first prepared detergent solubilized activated t-SNARE complex [see *Prinslow et al., 2019* by incubating 10 μM full-length syntaxin-1A with 15 μM synaptobrevin-2 (49–93), SNAP-25A(11-82) and SNAP-25A(141-203) in the presence of 1% β-OG, 0.5 M NaCl and protease inhibitor cocktail A overnight at 4˚C. For incorporation into nanodiscs, cisSNARE-complex, activated t-SNARE-complex or full-length synaptobrevin-2 were mixed with MSP1E3D1 and solubilized lipids at a ratio of 1:3:300 and incubated at 4˚C for 30 min. The detergent was removed using

Thermo Scientific Pierce Detergent Removal Resin, as described for the isolated nanodiscs, and size exclusion chromatography using a Superdex 200 column (GE 16/60). Fractions from size exclusion chromatography were assessed by SDS-PAGE and fractions that contained approximately one cisSNARE-complex, activated t-SNARE-complex or full-length synaptobrevin-2 per 2 MSP1E3D1 molecules were pooled together, mixed with protease inhibitor cocktail A and concentrated. TransSNARE-complex was formed by incubating nanodiscs containing activated t-SNARE complex with 2-fold excess of synaptobrevin nanodiscs overnight at 4˚C. This method leads to quantitative formation of trans-SNARE complexes between nanodiscs based on the complete release of the synaptobrevin-2 (49–93) peptide, which was assessed by labeling the peptide with $^{15}N$ and acquiring $^{1}H$-$^{15}N$ HSQC spectra, or labeling the peptide with Cy5 at residue 79 and monitoring the fluorescence anisotropy of the dye. For experiments comparing binding of $C_2AB*$ to cisSC-NDs and transSC-NDs, the cisSC-NDs were formed by incubating nanodiscs containing activated t-SNARE complex with 2-fold excess of synaptobrevin-2 (29–93) overnight at 4˚C.

To estimate the concentration of the resulting ND, cisSC-ND or transSC-ND samples, a stock solution of solubilized lipids containing 5% RhoPE was prepared as previously described in the Materials and methods section. Dilutions of this stock solution were made and the absorbance at 570 nm was measured to make a standard curve. Samples were solubilized in 1% b-OG and their absorbance at 570 nm was measured. This measurement and the standard curve were used to estimate the total lipids in nanodisc samples. The total lipid concentration was divided by 200 to get the approximate concentration of the nanodiscs.

To estimate the concentration of SNARE complex samples formed using SNAP-25(11–82) R16C, Q34C or K76C labeled with tetramethylrhodamine, the dye absorbance at 555 nm was used. For this purpose, the extinction coefficient of tetramethylrhodamine at 555 nm in 20 mM HEPES pH 7.4 125 mM KCl buffer was first determined from a standard curve obtained from the absorbance at 555 nm of several samples made after serial dilution of a stock solution of Tetramethylrhodamine-5-Maleimide.

## Assembly of soluble SNARE complex samples for FRET experiments

The SNARE motifs were mixed in an equimolar ratio in the following order: synaptobrevin-2 (29–93), SNAP-25A(141–203), SNAP-25A(11–82) K76C labeled with tetramethylrhodamine and syntaxin-1A (191–253) (for mcc) or syntaxin-1A (2–253) (for solubleSC), in the presence of 1 M NaCl. The assembly reaction was incubated in the presence of protease inhibitor cocktail A at room temperature overnight while rotating. The SNARE motifs that did not incorporate into the complex were removed by concentration-dilution at room temperature using 30 kDa Amicon centrifugation filters (*Chen et al., 2002*). The concentration of the SNARE complex was estimated by using UV-vis absorbance.

## FRET assay to monitor binding of Syt1 $C_2AB*$ to NDs, cisSC-NDs, transSC-NDs, solubleSC and mcc

A stock solution of 0.5 µM Alexa 488-tagged $C_2AB$ ($C_2AB*$), containing 5 µM BSA (Sigma Aldrich: A3059) to prevent binding of $C_2AB*$ to the cuvette, was prepared in the appropriate buffer (20 mM HEPES pH 7.4 and varying amounts of KCl and other components) on the day of the experiment. The ND, cisSC-ND, transSC-ND, solubleSC or mcc stock solutions were buffer exchanged into the proper buffer by using concentration-dilution or during the size exclusion chromatography step. Other components such as 1 mM $CaCl_2$, 1 mM EGTA, 2.5 mM $MgCl_2$, 2 mM ATP or a combination of these reagents were added after the concentration step. All the stock solutions were kept on ice and the buffers were kept at room temperature. The ND samples can be kept on ice and used for experiments during a period of three days in the presence of protease inhibitor cocktail A.

$C_2AB*$ fragments at 50 nM concentration were mixed with varying amount of NDs, cisSC-NDs or transSC-NDs in buffer containing 20 mM HEPES pH 7.4 and the concentrations of EGTA, $Ca^{2+}$, $Mg^{2+}$, ATP and KCl indicated in the figure legends. Each sample of a titration was prepared freshly by adding 20 µL of the stock $C_2AB*$ solution to an appropriate amount of ND, cisSC-ND or transSC-ND in a 1.5 mL Eppendorf tube. Dilutions of the ND stock solutions were used if less than 5 µL of the original stock had to be pipetted to minimize errors caused by pipetting small volumes. The volume was made up to 200 µL by adding buffer. The tube was vortexed for 2-3 s and 180 µL of the

solution was pipetted into an SOG cuvette (Fischer Scientific: NC9040289). Emission scan of the sample was immediately collected on a PTI Quantamaster 400 spectrofluorometer (T-format) at room temperature. All slits were set to 1.2 mm. The samples were excited at 480 nm and emission spectra from 500 to 620 nm were acquired. After the scan was collected, the sample was discarded, and the cuvette was washed with water followed by ethanol and dried using a stream of air. Multiple cuvettes were used during the experiment. This did not change the emission scans. Collecting the spectra after 5 min incubation at room temperature also did not change the emission scan. FRET efficiency was calculated using the formula: FRET = $(I_0 - I)/ I_0$, where $I_0$ is the fluorescence intensity of 50 nM Syt1 $C_2AB^*$ at 518 nm and I is the fluorescence intensity of the sample at 518 nm. The binding data were fitted to a Hill function $(FRET = Bmax^*(C\hat{}h)/((Kd\hat{}h) + (C\hat{}h))$, where C is the concentration of NDs, cisSC-NDs, transSC-NDs or solubleSCs, using GraphPad PRISM7.

## Statistics

The titrations of WT and mutant $C_2AB^*$ with NDs and cisSC-NDs under a variety of conditions and with different lipid compositions were performed over a long period of time as the project developed and the results suggested new experiments to be performed. The data shown in *Figures 5–9* and associated figure supplements, and summarized in *Supplementary file 1*, correspond to the last sets of experiments performed for presentation in this paper, all of which were consistent with the previous data. Each titration involving a given set of reagents and conditions was repeated at least twice and in most cases three or more times over the course of this work, but a comprehensive statistical analysis for groups of data corresponding to each figure panel is hindered by the fact that the groups of experiments performed previously as the project developed varied (e.g. with regard to which mutants were included). The reproducibility of the data that we obtained in this type of experiments is illustrated in *Figure 9—figure supplement 4* by the statistical analysis of the results of independent repeat experiments performed under the conditions of *Figure 6G,H* with the same set of mutants and three different preparations of NDs and cisSC-NDs. This analysis and, more importantly, the overall consistency of the data obtained in the large number of experiments that we performed strongly support the key conclusions drawn from the titrations.

## Acknowledgements

We thank Diana Tomchick, Axel Brunger and Thomas Südhof for fruitful discussions and helpful comments on the manuscript. The Agilent DD2 console of the 800 MHz spectrometer used for the research presented here was purchased with a shared instrumentation grant from the NIH (S10OD018027 to JR). Rashmi Voleti was supported by a fellowship from the Howard Hughes Medical Institute. This work was supported by grant I-1304 from the Welch Foundation (to JR) and by NIH Research Project Award R35 NS097333 (to JR).

## Additional information

### Funding

| Funder | Grant reference number | Author |
|---|---|---|
| National Institute of Neurological Disorders and Stroke | R35 NS097333 | Josep Rizo |
| Welch Foundation | I-1304 | Josep Rizo |
| Howard Hughes Medical Institute | | Rashmi Voleti |

The funders had no role in study design, data collection and interpretation, or the decision to submit the work for publication.

### Author contributions

Rashmi Voleti, Klaudia Jaczynska, Conceptualization, Formal analysis, Investigation, Methodology, Writing - review and editing; Josep Rizo, Conceptualization, Formal analysis, Funding acquisition, Investigation, Methodology, Writing - original draft

## Author ORCIDs

Josep Rizo  https://orcid.org/0000-0003-1773-8311

## Decision letter and Author response

Decision letter https://doi.org/10.7554/eLife.57154.sa1
Author response https://doi.org/10.7554/eLife.57154.sa2

## Additional files

### Supplementary files

• Supplementary file 1. Summary of apparent $K_D$s calculated from the titrations of $C_2AB^*$ with NDs, cisSC-NDs, transSC-NDs and solubleSCs performed under various conditions. Most of the entries in the table correspond to the data shown in *Figures 5–9* and associated figure supplements, as indicated in the second column. The first column shows whether the experiments were performed with WT $C_2AB^*$ or $C_2AB^*$ mutant E295A,Y338W (EAYW), R398Q,R399Q (RQRQ), R322E,K325E (REKE), K324E,K326E (KEKE) or R322E,K325E,R398Q,R399Q (REKERQRQ). The $K_D$s, $K_D$ errors, Hill coefficients (h) and FRET efficiencies at saturating concentrations (Bmax) were calculated from fitting the data to the Hill equation described in Materials and methods. Note that Bmax values above one must be considered an artifact of the fit, as FRET efficiencies cannot be larger than 1. The three columns on the right describe the buffer conditions used, the composition of the NDs, cisSC-NDs or transSC-NDs, and where the FRET acceptor probe was placed. SNN indicates the SNAP-25 N-terminal SNARE motif. To show the reproducibility of the data, the results obtained in three independent sets of experiments performed with different preparations under the conditions of *Figure 6G and H* are listed. Statistical analysis of these data is presented in *Figure 9—figure supplement 4*.

• Supplementary file 2. Cooperativity factors calculated as ratios of KDs measured in titrations of $C_2AB^*$ with: (i) NDs vs. cisSC-NDs; (ii) NDs or cisSC-NDs in EGTA in the absence vs. the presence of Mg-ATP; (iii) solubleSC vs. cisSC-NDs in the presence of EGTA or $Ca^{2+}$; and (iv) soluble SC vs. cisSC-PC-PS-PIP2-NDs in the presence of EGTA or $Ca^{2+}$. The Figures showing the data used to derive the cooperativity factors are indicated. Table Supplement 1 describes the KDs used for the calculations.

• Transparent reporting form

### Data availability

The NMR data corresponding to Figs. 1-4 and corresponding figure supplements are publicly available at https://doi.org/10.5061/dryad.0zpc866vt. Source data are provided for all the FRET experiments shown in Figs. 5-9 and corresponding figure supplements.

The following dataset was generated:

| Author(s) | Year | Dataset title | Dataset URL | Database and Identifier |
|---|---|---|---|---|
| Rizo J, Jaczynska K, Voleti R | 2020 | NMR data in 'Ca2+-dependent release of Synaptotagmin-1 from the SNARE complex on phosphatidylinositol 4,5-bisphosphatecontaining membranes' by Voleti, Jaczynska and Rizo, eLife 2020 | https://doi.org/10.5061/dryad.0zpc866vt | Dryad Digital Repository, 10.5061/dryad.0zpc866vt |

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
