## [Decision Letter]

**Acceptance summary:**

The manuscript represents a strong addition to the understanding of the functional interplay between SNAREs, membranes, synaptotagmin and, to a somewhat lesser extent perhaps, complexin in regulated secretion. The authors performed extensive NMR and fluorescence-based binding experiments with wild-type and informative mutant proteins, both in soluble form and (for the SNAREs) embedded in nanodiscs of various compositions. This manuscript will likely be seen as a landmark contribution to the ever-improving understanding of the remarkably complex system of regulated membrane fusion.

**Decision letter after peer review:**

Thank you for submitting your article "Dissection of Synaptotagmin-1-SNARE complex interactions in solution and on membranes" for consideration by *eLife*. Your article has been reviewed by Olga Boudker as the Senior Editor, a Reviewing Editor, and three reviewers. The following individuals involved in review of your submission have agreed to reveal their identity: Angel Perez Lara (Reviewer #2).

The reviewers and I have discussed the reviews with one another and I as Reviewing Editor have drafted this decision to help you prepare a revised submission.

As the editors have judged that your manuscript is of interest, but as described below that additional experiments are required before it is published, we would like to draw your attention to changes in our revision policy that we have made in response to COVID-19 (https://elifesciences.org/articles/57162). First, because many researchers have temporarily lost access to the labs, we will give authors as much time as they need to submit revised manuscripts. We are also offering, if you choose, to post the manuscript to bioRxiv (if it is not already there) along with this decision letter and a formal designation that the manuscript is 'in revision at *eLife*'. Please let us know if you would like to pursue this option. (If your work is more suitable for medRxiv, you will need to post the preprint yourself, as the mechanisms for us to do so are still in development.)

All three reviewers and I agree that your paper is a very good *eLife* candidate. Our consensus is that your work represents a strong addition to our understanding of the synaptic vesicle fusion process by describing important aspects of the functional interplay between SNAREs, membranes, synaptotagmin, and complexin.

At the same time, the reviewers commented on a series of issues that we think need to be dealt with before your work can be published in *eLife*. The corresponding comments concern a set of more conceptual, interpretational, and methodological aspects, which can mostly be dealt with by careful revision of the manuscript text (A below), and include suggestions for a set of additional experiments and data analyses that we think are needed to support some of the key conclusions (B below).

Essential Revisions:

A) Conceptual, Interpretational, and Methodological Aspects:

i) While Zhou et al., (2017) used Cpx1-1-83 (crystallography) and Cpx1-48-73 (ITC), the present study employed Cpx1-26-83. This issue needs to be discussed as it may be problematic to compare characteristics of complexes formed with these different Cpx1 constructs. In essence it is possible that the type of Cpx1 fragment contributes to the relative stability and dynamics of the tripartite versus primary interface. Unless comparative experiments with exactly the same constructs are done, it is difficult to resolve discrepancies between the present and previous studies.

ii) Related to the point above (i), the 'assault' on the tripartite complex published by Zhou et al., (2017) appears overdone in some places and not always fully supported by the data. Zhou et al., (2017) really only observed the impact of the tripartite perturbations (LLQQ mainly) when they mutated away the primary interface using the quintuple C_2_B and quintuple SNAP25. In the present study the equivalent experiment was not done, and different protein fragments were used, so it is not clear if similar observations would have been made with NMR if exactly the same conditions/fragments would have been used. This issue should be taken into account when discussing the study by Zhou et al., (2017).

iii) The present study claims that LLQQ destabilizes Syt1, which then explains the published impact of this variant on synaptic transmission. This should be documented in the evidence as it is a crucial and impactful finding.

iv) Solution-state NMR is closer to physiologically significant conditions than crystallography. However, there are still major unresolved discrepancies between the present data and the ITC data of Zhou et al., (2017), where the experimental conditions are more comparable. In view of this and some other aspects, the 'physiological accuracy' of the experimental conditions in the present study appears oversold.

v) The discussion of the nanodisc-based data should consider the fact that clusters of SNAREs or Syt1 cannot form under these conditions even though these clusters do/may exist under physiological conditions and thus impose specific constraints on the prefusion complex and its relative stability. In essence, there is a possibility that physiologically important complexes are missed in the nanodisc scenario.

vi) In the study by Zhou et al., (2015), the destruction of the primary interface eliminated Syt1 function for both Ca^2+^-triggered fusion and inhibition of spontaneous release (clamping function), while the REKE and EAYW both strongly impaired Ca^2+^-triggered fusion without disrupting clamping. These perturbations occur on opposite sides of the β sandwich pointing in opposite directions and yet they cause very similar phenotypes at the synapse. Do both perturbations stabilize the primary interface at the expense of some other competitive interaction required for breaking the clamp upon Ca^2+^ binding? Or is there another interface not utilized during clamping but important for fusion that both these regions could contribute to? None of the three model interfaces (polybasic, primary, tripartite) simultaneously incorporate the RK and EY residue pairs at a shared interface. The authors of the present study should elaborate on the effects of REKE and EAYW in this context, perhaps by clarifying their own thinking on the issue.

B) Additional Data Analyses and Experiments:

i) The interpretation of NMR results is based on crystal structures. ITC experiments indicate a SNARE:C_2_B ratio of 2:1 – thus some of the results could be explained by cooperativity effects instead of competition between different binding sites. This should be analysed/discussed.

ii) In order to properly compare the Syt1 RQRQ, REKE, KEKE, and REKERQRQ variants, cooperativity factors need to be determined in order to assess the contribution of SNARE binding vs. membrane binding. Corresponding analyses of the WT, REKE, and RQRQ variants would suffice.

iii) To distinguish effects of variants on SNARE vs. membrane binding, additional experiments in the absence of anionic phospholipids are important. For instance, SNAP-25 could be used to monitor Syt11 C_2_AB binding (RQRQ, REKE, KEKE and REKERQRQ) to SNAREs using nanodisc with or without PS and PIP_2_, and with and without calcium. This would provide insight into the relative contribution of SNARE binding vs. membrane binding. Corresponding analyses of the WT, REKE, and RQRQ variants would suffice.

iv) Cooperativity factors (K_D_ cisSC-ND/K_D_ ND, K_D_ cisSC-ND/K_D_ cisSC-ND, K_D_ ND/K_D_ ND ratios) in the presence and absence of ATP/magnesium need to be determined, compared, and discussed.

[Editors' note: further revisions were suggested, as described below.]

Reviewer #1:

I continue to be enthusiastic about this manuscript, which I think is substantially above the bar for publishing in eLife. For reference, I am including below my original, positive review (slightly edited).

This manuscript represents a strong addition to our understanding of the functional interplay between SNAREs, membranes, synaptotagmin and, to a somewhat lesser extent perhaps, complexin. The authors have performed extensive NMR and fluorescence-based binding experiments with wild-type and informative mutant proteins, both in soluble form and (for the SNAREs) embedded in nanodiscs of various compositions. Although at first I found the sheer length of the manuscript daunting, I became convinced as I read it that the length is justified by the wealth of data and the need to carefully integrate the results with the extensive existing literature. I think the authors have succeeded to an impressive degree, and that this manuscript will be seen as a landmark contribution to our ever-improving understanding of this remarkably complex system.

There are many, many interesting results, but I would point single out three as deserving of special mention:

1. The striking differences, in vivo and in vitro, between two different double mutants (R322E/K325E versus K324E/K326E) in the same portion of synaptotagmin's polybasic region. The former has a much stronger effect that the latter on both PIP2-dependent membrane binding (this work) and neurotransmitter release (Brewer 2015). This can be beautifully reconciled by a detailed consideration of how the calcium-binding loops and the polybasic region cooperate in membrane binding of Ca2+-bound synaptotagmin (see Fig. 10B) and thereby promote fusion.

2. The remarkable finding that the E295A/Y338W mutation in the primary interface, previously claimed to abrogate synaptotagmin-SNARE binding (Zhou 2015), actually strengthens it. Given that this mutation compromises neurotransmitter release, the results highlight that overly avid binding can be as deleterious as compromised binding in vivo. I'm not sure why the Brunger lab reached the opposite conclusion about the impact of this double mutation on in vitro binding (Zhou 2015), but I found the results presented here to be convincing.

3. The strong implication that the tripartite SNARE:synaptotagmin:complexin interface, which was the main point of the Brunger lab's Nature paper and supported by ITC analysis of wild-type and mutant proteins (Zhou 2017), is a nonetheless a crystallization artifact, possibly induced by the artificial tether used in that work to link synaptotagmin to the SNARE bundle. Further work is underway, in collaboration with the Brunger lab, to resolve the contradiction, but my hunch is that the tripartite interface may not actually be physiologically important.

This manuscript represents an enormous amount of careful work, complemented by a highly intelligent and generally quite balanced synthesis of previous data from the author's and other labs. As such, I believe that it will be viewed as essential reading for anyone interested in the question of how, at a molecular mechanistic level, synaptic vesicles fuse in response to calcium.

*Reviewer #2*:

The authors have addressed my main concerns. They have carried out the requested experiments in order to clarify and support their conclusions. They have re-analyzed the results and re-written the manuscript regarding to the new data analysis and results. However, I still have one major concern:

I agree with authors that the stoichiometry for the binding of the SNARE complex to the C2B domain has not been reported but, as we can see in figure 3 b and d from Zhou Q et al., ITC results suggest a number of binding site (N) around 0.5 , what means 2 SNARE complex: 1 C2B domain stoichiometry. Additionally, this binding mode was suggested in the Extended Data Figure 9 of the same paper. Furthermore, ITC results argue against 1 C2B domain per SNARE complex, taking into account that the plateau starts around 1 molar ratio.

The authors claim "... that there is an interplay between binding of CpxSC to the primary interface and to the polybasic region. CpxSC appears to bind exclusively to one or the other site at low concentrations, likely because of steric hindrance disfavors simultaneous binding of two CpxSC complexes to one C2B domain. However, simultaneous binding to both sites might be allowed at higher CpxSC concentrations by slight alterations in both binding modes, leading to the curved cross-peak movement.'

I have some concerns about this issue. Taking into account the design of the ITC experiment (see above), at the very beginning we would get 2:1 stoichiometry for binding of the SNARE complex to the C2B domain because of there is a high concentration of complex against the low concentration of C2 domain. However, as we are injecting C2 domain, in a competition scenario, the experiment should present a number of binding site of 1 instead around 0.5, because of the increase of the C2 domain concentration will decrease the CpxSC:C2 domain ratio and , as consequence, several C2 domains will be accessible for competition per CpxSC complex. In addition, we can observe a chemical shift at low concentrations in the polybasic region (K327, K369 and K325) in the EAYW mutant (Fig. 4 C of the manuscript). Could it be possible a scenario in which two CpxSC complexes bind to 1 C2 domain ( even at low concentration of CpxSC complex) and the different mutations tune the interplay of the polybasic region and primary interface ( maybe even promoting different binding modes), giving the observed NMR results? We have to take into account that the binding is dynamic (as reported previously Brewer et al., 2015).

Despite of the above comments, I am aware of the effort carried out by the authors and in my opinion they addressed my main concerns and I suggest and support the publication of the manuscript in eLife.

Reviewer #3:

The authors have dealt with my major scientific concerns (comparisons of identical CPX constructs with Brunger lab results) and clarified their conclusions/interpretations of the EAYW vs REKE mutations. They have also convincingly demonstrated that the tripartite binding mode does not contribute to SNARE interactions under their experimental conditions while also toning down their conclusion that the tripartite mode is entirely a crystal artifact. I agree with the authors that the specific Brunger lab comparisons should be published separately and in coordination with that lab rather than included here. The edits made the manuscript a bit easier to read and I appreciate the authors trying to limit the amount of new data included to avoid expanding the text/figures even more. The discussion section is still quite hefty, and I would recommend some more editing for clarity, but this is a minor issue. Overall, this is an impressive amount of data and the conclusions contain some important new insights. I support going forward with publication.

---

## [Author Response]

Essential Revisions:A) Conceptual, Interpretational, and Methodological Aspects:i) While Zhou et al., (2017) used Cpx1-1-83 (crystallography) and Cpx1-48-73 (ITC), the present study employed Cpx1-26-83. This issue needs to be discussed as it may be problematic to compare characteristics of complexes formed with these different Cpx1 constructs. In essence it is possible that the type of Cpx1 fragment contributes to the relative stability and dynamics of the tripartite versus primary interface. Unless comparative experiments with exactly the same constructs are done, it is difficult to resolve discrepancies between the present and previous studies.

This is a valid point, but it is important to note that the difference between Cpx1-26-83 used in our study and Cpx1-1-83 used for crystallography is at the Cpx1 N-terminus, which is very far from the tripartite interface and hence is not expected to affect binding. Note also that Cpx1-48-73 used for ITC is truncated near the putative binding interface and hence binding results obtained with longer fragments such as Cpx1-26-83, which do not lack residues near the interface, are expected to be more reliable. In fact, Cpx126-83 binds with substantially higher affinity to the SNARE complex than Cpxl1-48-73 (Xu et al., 2013; Zhou et al., 2017).

ii) Related to the point above (i), the 'assault' on the tripartite complex published by Zhou et al., (2017) appears overdone in some places and not always fully supported by the data. Zhou et al., (2017) really only observed the impact of the tripartite perturbations (LLQQ mainly) when they mutated away the primary interface using the quintuple C_2_B and quintuple SNAP25. In the present study the equivalent experiment was not done and different protein fragments were used, so it is not clear if similar observations would have been made with NMR if exactly the same conditions/fragments would have been used. This issue should be taken into account when discussing the study by Zhou et al., (2017).

We agree that we should tone down the ‘assault’ on the tripartite complex, even though we believe that the data shown in the paper already provide very strong evidence that the tripartite interface is not substantially populated at the concentrations used in the NMR experiments. Thus, we have shortened the paragraph describing the conflict between our NMR data and the ITC data, and have softened the tone. The paragraph now reads (subsection “Mutation of R322,K325 at the polybasic region and R398,R399 at the primary interface abolishes C_2_B SNARE complex binding in solution”):

“These results contrast with ITC data suggesting the existence of a Syt1-complexin-1SNARE complex interaction involving the tripartite interface in experiments performed with Syt1 C_2_B bearing seven mutations designed to disrupt binding via the polybasic region and the primary interface (KA-Q mutant), and a complexin-1-SNARE complex bearing five mutations to further disrupt such binding (Zhou et al., 2017). We are collaborating with the laboratory of Axel Brunger to determine the reasons underlying these conflicting results, and the results of these efforts will be published elsewhere.”

We have also shorted and softened the summary paragraph where in the original manuscript we were strongly suggesting that the tripartite complex arises from crystal contacts, which now reads (subsection “Mutation of R322,K325 at the polybasic region and R398,R399 at the primary interface abolishes C_2_B SNARE complex binding in solution”):

“In summary, our data show that binding of the Syt1 C_2_B domain to the Cpx1(26-83)SNARE complex in solution involves the polybasic region and the primary interface, while binding via the tripartite interface is undetectable under the conditions of our NMR experiments even when CpxSC is added at 85 μM concentration and binding through the polybasic region and the tripartite interface is abolished. These results indicate that the tripartite interface observed in the complexin-1-Syt1-SNARE complex crystals might have arisen from crystal packing, but further research will be required to clarify this issue.”

We also note that, in the last paragraph of the Discussion section of the original manuscript, we left open the possibility that the tripartite complex might be physiologically relevant, although reminded the reader that ‘the instability caused by the L387Q/L394Q mutation might underlie the phenotypes caused by this mutation, rather than disruption of the tertiary interface’. We have removed this reminder and the relevant part of this paragraph now reads:

“Note also that, while our NMR studies could not detect binding of the Syt1 C_2_B domain to CpxSC through the tripartite interface (Figure 2E, Figure 3), it is premature to completely rule out the relevance of the tripartite structure given the potential functional importance of very weak interactions (Magdziarek et al., 2020). Nevertheless, it is also premature to accept this structure as physiologically relevant, and further evidence for such relevance needs to be obtained, ideally using mutations that exclusively replace residues in the interface rather than interior residues that are important for protein stability.”

iii) The present study claims that LLQQ destabilizes Syt1, which then explains the published impact of this variant on synaptic transmission. This should be documented in the evidence as it is a crucial and impactful finding.

We made this claim based on the NMR data shown in Figure 4 of the original manuscript, on the observation that this mutant has much higher tendency to precipitate with CpxSC and, most of all, on the thermal denaturation data described by Zhou et al., 2017 (Extended data Figure 3). To make the latter point more clear, we now describe the Zhou et al., 2017 data in a little more detail in the sentence (subsection “Mutation of R322,K325 at the polybasic region and R398,R399 at the primary interface abolishes C_2_B SNARE complex binding in solution”):

“These results show that the L387Q/L394Q causes a considerable destabilization of the Syt1 C_2_B domain, which is consistent with the finding that this mutation decreases the thermal denaturation temperature of the C_2_B domain by about 10 ^o^C (Zhou et al., 2017).”

In the spirit of tempering the ‘assault’ on the tripartite complex (point ii above), in the revised manuscript we have moved the NMR data of the old Figure 4 to a supplementary figure (Figure 3—figure supplement 1), and we have moved the sentence suggesting that the phenotypes caused by the L387Q/L394Q mutation arise from protein destabilization from the summary paragraph (subsection “Mutation of R322,K325 at the polybasic region and R398,R399 at the primary interface abolishes C_2_B SNARE complex binding in solution”).

iv) Solution-state NMR is closer to physiologically significant conditions than crystallography. However, there are still major unresolved discrepancies between the present data and the ITC data of Zhou et al. (2017), where the experimental conditions are more comparable. In view of this and some other aspects, the 'physiological accuracy' of the experimental conditions in the present study appears oversold.

We are not sure which parts of the original manuscript led to this concern, as we had pointed out explicitly the limitations of our own data in multiple places of the manuscript (e.g. lack of membranes in our NMR experiments and the potential contributions of non-specific interactions to the results obtained with nanodiscs). In any case, we imagine that the paragraphs that we have modified to address concerns ii-iii above contributed to concern iv. We hope that we have properly addressed this concern with the modifications introduced in those paragraphs.

v) The discussion of the nanodisc-based data should consider the fact that clusters of SNAREs or Syt1 cannot form under these conditions even though these clusters do/may exist under physiological conditions and thus impose specific constraints on the prefusion complex and its relative stability. In essence, there is a possibility that physiologically important complexes are missed in the nanodisc scenario.

We had mentioned the oligomers at the end of the Discussion section of the original manuscript and we now include the possibility of cluster formation in the same paragraph of the revised manuscript:

“Thus, syntaxin-1 is known to form clusters on the plasma membrane (Khuong et al., 2013), and formation of Syt1 oligomers has been proposed to underlie the primed state and to exert an inhibitory activity that is released by Ca^2+^ (Wang et al., 2017). Such clusters and oligomers could not be present in our nanodisc experiments. However, it is worth noting that the Syt1 oligomers are compatible with the state proposed in Figure 10A (Tagliatti et al., 2020) and hence can be readily incorporated into the model of Figure 10C.”

vi) In the study by Zhou et al., (2015), the destruction of the primary interface eliminated Syt1 function for both Ca^2+^-triggered fusion and inhibition of spontaneous release (clamping function), while the REKE and EAYW both strongly impaired Ca^2+^-triggered fusion without disrupting clamping. These perturbations occur on opposite sides of the β sandwich pointing in opposite directions and yet they cause very similar phenotypes at the synapse. Do both perturbations stabilize the primary interface at the expense of some other competitive interaction required for breaking the clamp upon Ca^2+^ binding? Or is there another interface not utilized during clamping but important for fusion that both these regions could contribute to? None of the three model interfaces (polybasic, primary, tripartite) simultaneously incorporate the RK and EY residue pairs at a shared interface. The authors of the present study should elaborate on the effects of REKE and EAYW in this context, perhaps by clarifying their own thinking on the issue.

This is a very good point. Our data suggest that the EAYW and REKE mutations give similar phenotypes because they both hinder release of the Syt1-SNARE complex interactions, the former by strengthening the interaction, the latter by hindering the specific Ca^2+^- and PIP_2_-dependent interaction of the C_2_B domain with the membrane that releases Syt1-SNARE complex binding. Release of this binding is likely to be necessary not only for evoked release but also for spontaneous release. Basically, we believe that the two mutations represent ‘two sides of the same coin’ much as the mutated residues are on two sides of the β-sandwich. In the revised manuscript we have added the following paragraph (Discussion section):

“The E295A/Y338W mutation in the primary interface also disrupted neurotransmitter release but still allowed clamping of spontaneous release, which resembles the phenotype observed for the R322E/K325E mutation in the polybasic region (Zhou et al., 2015). […] Similarly, the R322E/K325E mutation impairs Ca^2+^- and PIP_2_dependent membrane binding, which is critical to release this inhibitory interaction.”

B) Additional Data Analyses and Experiments:i) The interpretation of NMR results is based on crystal structures. ITC experiments indicate a SNARE:C_2_B ratio of 2:1 – thus some of the results could be explained by cooperativity effects instead of competition between different binding sites. This should be analysed/discussed.

We are not aware of any published ITC data that definitively demonstrate a 2:1 stoichiometry for binding of the SNARE complex to the C_2_B domain, even if the data might suggest this stoichiometry. In any case, our NMR results are compatible with such stoichiometry, but argue against cooperativity. The NMR results are interpreted based in part on the three structures available (one elucidated by NMR spectroscopy and two by X-ray crystallography), but also on the observation of the behaviors of the ^1^H-^15^N HSQC cross-peaks observed in the titrations. As discussed in the paper (subsection “Mutation of R322,K325 at the polybasic region and R398,R399 at the primary interface abolishes C2B SNARE complex binding in solution” and subsection “The E295A/Y338W mutation in the primary interface enhances C2B-CpxSC binding in solution”), comparison of the titrations performed with the WT C_2_B domain and the R322E/K325E and E295A/Y338W mutants show clear evidence of competition between binding of CpxSc to the polybasic and primary interfaces of the WT C_2_B domain at low CpxSC concentrations. However, the curved behavior of some cross-peaks at high CpxSc concentrations suggest that simultaneous binding to both interfaces might occur with slight alterations of the binding modes due to steric hindrance between the two bound CpxSCs. We have tried to explain better this conclusion rephrasing a few sentences (subsection “The E295A/Y338W mutation in the primary interface enhances C_2_B-CpxSC binding in solution”):

“This finding suggests that there is an interplay between binding of CpxSC to the primary interface and to the polybasic region. CpxSC appears to bind exclusively to one or the other site at low concentrations, likely because of steric hindrance disfavors simultaneous binding of two CpxSC complexes to one C_2_B domain. However, simultaneous binding to both sites might be allowed at higher CpxSC concentrations by slight alterations in both binding modes, leading to the curved cross-peak movement.”

ii) In order to properly compare the Syt1 RQRQ, REKE, KEKE, and REKERQRQ variants, cooperativity factors need to be determined in order to assess the contribution of SNARE binding vs. membrane binding. Corresponding analyses of the WT, REKE, and RQRQ variants would suffice.

Supplementary file 2 now describes cooperativity factors obtained by dividing the K_D_s observed with NDs by the K_D_s observed with cisSC-NDs.

iii) To distinguish effects of variants on SNARE vs. membrane binding, additional experiments in the absence of anionic phospholipids are important. For instance, SNAP-25 could be used to monitor Syt11 C_2_AB binding (RQRQ, REKE, KEKE and REKERQRQ) to SNAREs using nanodisc with or without PS and PIP2, and with and without calcium. This would provide insight into the relative contribution of SNARE binding vs. membrane binding. Corresponding analyses of the WT, REKE, and RQRQ variants would suffice.

We have performed new titrations of C_2_AB* with cisSC-NDs that contained only PC or 84% PC, 15% PS and 1% PIP_2_, and that were labeled with an acceptor TMR probe on residue 76 of SNAP-25. This design is expected to yield high FRET efficiency for binding of C_2_AB* to the SNARE complex via the primary or polybasic interfaces. We included WT, REKE and RQRQ C_2_AB* in these experiments, as suggested by the reviewers, and also the EAYW C_2_AB* mutant that enhances SNARE binding. We compared the results with parallel titrations performed with soluble SNARE complex formed with the cytoplasmic region of syntaxin-1 (for comparison with the cisSC-ND data) or with only the syntaxin-1 SNARE motif (for comparison with the NMR data and to assess the potential contribution of the syntaxin-1 N-terminal region to binding). The results are presented in the new Figure 8 and Figure 8—figure supplement 1.

The results generally confirmed the conclusions obtained with our previous FRET experiments and, importantly, yielded a striking result: while most of the experiments revealed high FRET efficiency at high SNARE complex concentrations, the maximum FRET efficiency was about 0.4 for titrations of WT C_2_AB* with cisSC-PC/PS/PIP_2_-NDs, even though the affinity was higher than under all other conditions. This result and the effects of the mutations on the affinity and maximum FRET efficiency (Figure 8H) strongly support the conclusion that Ca^2+^-dependent binding of C_2_AB* to PIP_2_-containing membranes competes with the Syt1-SNARE complex interaction, as explained in more detailed in the new section describing these data (subsection “The E295A/Y338W mutation in the primary interface enhances Ca^2+^-independent binding of C2AB* to cisSNARE complex-nanodiscs”). Models of how the Syt1 C_2_B domain can interact with the SNARE complex and the nanodiscs are now shown in Figure 5—figure supplement 2 to help rationalizing these data and, more generally, the results of the FRET assays shown in Figure 5, Figure 6, Figure 7, Figure 8, Figure 9. In addition, in Supplementary file 2 we describe cooperativity factors calculated by dividing the K_D_s measured with soluble SNARE complexes by the K_D_s measured with cisSC-NDs.

iv) Cooperativity factors (K_D_ cisSC-ND/K_D_ ND, K_D_ cisSC-ND/K_D_ cisSC-ND, K_D_ ND/K_D_ ND ratios) in the presence and absence of ATP/magnesium need to be determined, compared, and discussed.

In Supplementary file 2 we now describe cooperativity factors calculated by dividing K_D_s measured in the presence of Mg-ATP by K_D_s measured in the absence of Mg-ATP. However, to avoid lengthening of the manuscript, we limit the discussion of cooperativity factors to places where these factors are particularly informative (e.g. subsection “Mutations in the polybasic region and R398,R399 disrupt Ca^2+^-independent binding of C_2_AB* to nanodisc-anchored SNARE complex”, subsection “Ca^2+^- and PIP_2_-dependent binding of the C_2_B domain to membranes hinders SNARE complex binding” and subsection “Effects of phospholipids on C_2_AB* binding to the SNARE complex”).

[Editors' note: further revisions were suggested, as described below.]

Reviewer #1:I continue to be enthusiastic about this manuscript, which I think is substantially above the bar for publishing in eLife. For reference, I am including below my original, positive review (slightly edited).This manuscript represents a strong addition to our understanding of the functional interplay between SNAREs, membranes, synaptotagmin and, to a somewhat lesser extent perhaps, complexin. The authors have performed extensive NMR and fluorescence-based binding experiments with wild-type and informative mutant proteins, both in soluble form and (for the SNAREs) embedded in nanodiscs of various compositions. Although at first I found the sheer length of the manuscript daunting, I became convinced as I read it that the length is justified by the wealth of data and the need to carefully integrate the results with the extensive existing literature. I think the authors have succeeded to an impressive degree, and that this manuscript will be seen as a landmark contribution to our ever-improving understanding of this remarkably complex system.There are many, many interesting results, but I would point single out three as deserving of special mention:1. The striking differences, in vivo and in vitro, between two different double mutants (R322E/K325E versus K324E/K326E) in the same portion of synaptotagmin's polybasic region. The former has a much stronger effect that the latter on both PIP2-dependent membrane binding (this work) and neurotransmitter release (Brewer 2015). This can be beautifully reconciled by a detailed consideration of how the calcium-binding loops and the polybasic region cooperate in membrane binding of Ca2+-bound synaptotagmin (see Fig. 10B) and thereby promote fusion.2. The remarkable finding that the E295A/Y338W mutation in the primary interface, previously claimed to abrogate synaptotagmin-SNARE binding (Zhou 2015), actually strengthens it. Given that this mutation compromises neurotransmitter release, the results highlight that overly avid binding can be as deleterious as compromised binding in vivo. I'm not sure why the Brunger lab reached the opposite conclusion about the impact of this double mutation on in vitro binding (Zhou 2015), but I found the results presented here to be convincing.3. The strong implication that the tripartite SNARE:synaptotagmin:complexin interface, which was the main point of the Brunger lab's Nature paper and supported by ITC analysis of wild-type and mutant proteins (Zhou 2017), is a nonetheless a crystallization artifact, possibly induced by the artificial tether used in that work to link synaptotagmin to the SNARE bundle. Further work is underway, in collaboration with the Brunger lab, to resolve the contradiction, but my hunch is that the tripartite interface may not actually be physiologically important.This manuscript represents an enormous amount of careful work, complemented by a highly intelligent and generally quite balanced synthesis of previous data from the author's and other labs. As such, I believe that it will be viewed as essential reading for anyone interested in the question of how, at a molecular mechanistic level, synaptic vesicles fuse in response to calcium.

We are very thankful for the kind comments and for the nice distillation of the most important among our results. We could not agree more.

Reviewer #2:The authors have addressed my main concerns. They have carried out the requested experiments in order to clarify and support their conclusions. They have re-analyzed the results and re-written the manuscript regarding to the new data analysis and results. However, I still have one major concern:I agree with authors that the stoichiometry for the binding of the SNARE complex to the C2B domain has not been reported but, as we can see in figure 3 b and d from Zhou Q et al., ITC results suggest a number of binding site (N) around 0.5 , what means 2 SNARE complex: 1 C2B domain stoichiometry. Additionally, this binding mode was suggested in the Extended Data Figure 9 of the same paper. Furthermore, ITC results argue against 1 C2B domain per SNARE complex, taking into account that the plateau starts around 1 molar ratio.The authors claim "... that there is an interplay between binding of CpxSC to the primary interface and to the polybasic region. CpxSC appears to bind exclusively to one or the other site at low concentrations, likely because of steric hindrance disfavors simultaneous binding of two CpxSC complexes to one C2B domain. However, simultaneous binding to both sites might be allowed at higher CpxSC concentrations by slight alterations in both binding modes, leading to the curved cross-peak movement.'I have some concerns about this issue. Taking into account the design of the ITC experiment (see above), at the very beginning we would get 2:1 stoichiometry for binding of the SNARE complex to the C2B domain because of there is a high concentration of complex against the low concentration of C2 domain. However, as we are injecting C2 domain, in a competition scenario, the experiment should present a number of binding site of 1 instead around 0.5, because of the increase of the C2 domain concentration will decrease the CpxSC:C2 domain ratio and , as consequence, several C2 domains will be accessible for competition per CpxSC complex. In addition, we can observe a chemical shift at low concentrations in the polybasic region (K327, K369 and K325) in the EAYW mutant (Fig. 4 C of the manuscript).Could it be possible a scenario in which two CpxSC complexes bind to 1 C2 domain ( even at low concentration of CpxSC complex) and the different mutations tune the interplay of the polybasic region and primary interface ( maybe even promoting different binding modes), giving the observed NMR results? We have to take into account that the binding is dynamic (as reported previously Brewer et al., 2015).

We respectfully believe that the reviewer over-interprets the ITC data of Zhou et al. 2017. ITC is a powerful technique to quantify the affinity of binding interactions when strong heats and nicely sigmoidal curves are observed, but is much less informative when heats are smaller and the curves are not clearly sigmoidal. Moreover, analysis of this kind of data is further complicated when there is more than one binding mode. Hence, while it is true that the black curve in Fig. 3b of Zhou et al. could be explained by a 2:1 stoichiometry of binding (N = 0.5), this conclusion is far from definitive. For instance, the data could also arise if an endothermic binding mode dominates at the beginning of the titration and the endothermic signal is progressively cancelled by the signal from a weaker exothermic interaction at higher C2B concentrations. There may also be factors that contribute to weak ITC signals and are difficult to account for. For instance, the data of Fig. 3d mentioned by the reviewer is also compatible with N = 0.5, but these data were acquired with the mutant proteins where presumably binding through the polybasic and primary interfaces was abolished; thus, it is highly unclear how a 2:1 complex could be formed.

With respect to the NMR data mentioned by the reviewer, the scenario is certainly possible, but the proposed 2:1 complex does not appear to be the dominant species, as the cross-peak shifts observed for the residues of the polybasic region occur more slowly than for the WT C2B domain.

Despite of the above comments, I am aware of the effort carried out by the authors and in my opinion they addressed my main concerns and I suggest and support the publication of the manuscript in eLife.

We very much appreciate the kind comments and the multiple constructive comments from the reviewer, which have helped us to substantially improve the quality of the manuscript.

Reviewer #3:The authors have dealt with my major scientific concerns (comparisons of identical CPX constructs with Brunger lab results) and clarified their conclusions/interpretations of the EAYW vs REKE mutations. They have also convincingly demonstrated that the tripartite binding mode does not contribute to SNARE interactions under their experimental conditions while also toning down their conclusion that the tripartite mode is entirely a crystal artifact. I agree with the authors that the specific Brunger lab comparisons should be published separately and in coordination with that lab rather than included here. The edits made the manuscript a bit easier to read and I appreciate the authors trying to limit the amount of new data included to avoid expanding the text/figures even more. The discussion section is still quite hefty, and I would recommend some more editing for clarity, but this is a minor issue. Overall, this is an impressive amount of data and the conclusions contain some important new insights. I support going forward with publication.

We also thank the reviewer for the kind comments and for the constructive criticisms. We have revised some parts of the discussion to improve clarity and have shortened the discussion a little while making every effort to avoid sacrificing clarity. As a result, the discussion is now almost one page shorter.